# Analysis of ozone vertical profiles day-to-day variability in the lower troposphere during the Paris-2022 ACROSS campaign

Gérard Ancellet[1], Camille Viatte[1], Anne Boynard[1,2], François Ravetta[1], Jacques Pelon[1], Cristelle Cailteau-Fischbach[1], Pascal Genau[1], Julie Capo[3], Axel Roy[3], and Philippe Nédélec[4]

[1]LATMOS, Sorbonne Université, Université Versailles St-Quentin, CNRS/INSU, Paris, France
[3]CNRM, Université de Toulouse, Météo-France, CNRS, Toulouse, France
[4]Laboratoire d'Aérologie, CNRS and Université Toulouse III, Paul Sabatier, Toulouse, France
[2]SPASCIA, Ramonville-Saint-Agne, 31520, France

**Correspondence:** Gérard Ancellet (gerard.ancellet@latmos.ipsl.fr)

**Abstract.** The $O_3$ vertical profiles variability in the lower troposphere is analyzed during the summer 2022 ACROSS (Atmospheric ChemistRy Of the Suburban foreSt) measurement campaign as part of the PANAME (PAris region urbaN Atmospheric observations and models for Multidisciplinary rEsearch) project. The analysis is based on 21 days of DIfferential Absorption Lidar (DIAL) observations, in addition to the two daily vertical $O_3$ profiles measured by In-service Aircraft for a Global Observing System (IAGOS) flights to and from Paris airport. The ACROSS $O_3$ profiles are also a good opportunity to assess the lowermost tropospheric $O_3$ column retrieval by the satellite observations of Infrared Atmospheric Sounding Interferometer (IASI). The planetary boundary layer (PBL) vertical structure evolution is monitored using a 808-nm microlidar and meteorological radiosondes launched in the city center. Characterization of the regional transport of polluted air masses advected over the city is based on the daily $O_3$ analysis of the Copernicus Atmospheric Service (CAMS) ensemble model and on backward trajectories of the Paris city plume. This work show that the CAMS simulations of the Paris $O_3$ plume between the surface and 3 km are consistent with the ACROSS $O_3$ vertical profiles and that the AM IASI satellite observations can capture the day to day variability of the 0-3 km lowermost $O_3$ column only when the maximum altitude of the lower tropospheric ozone plume is higher than 2 km. The day time $O_3$ vertical structure above the city center is also in good agreement with the PBL growth during the day and with the formation of the residual layer during the night. The $O_3$ DIAL may provide additional information about the PBL vertical structure to discuss differences between microlidar and radiosounding measurements of the PBL height.

In addition to the well-known control of the $O_3$ photochemical production by atmospheric temperature, cloud cover and mixing between the surface layer (0 - 500 m) and the residual layer, the comparison of four $O_3$ pollution events shows that the thickness of the PBL during the day and the advection of regional scale plumes above the PBL can significantly change the $O_3$ concentrations above Paris city center. With similar cloud cover and air temperature, high $O_3$ concentrations up to 180 µg.m$^{-3}$ are encountered during the day when PBL height is below 1.5 km, while they remain below 150 µg.m$^{-3}$ when PBL height increases above 2.5 km. Advection of $O_3$ poor concentrations in the free troposphere during a Saharan dust event is able to mitigate the $O_3$ photochemical production. On the other hand, the advection of a pollution plume from continental Europe with high $O_3$ concentrations $> 140$ µg.m$^{-3}$ maintained high concentrations in the surface layer despite a temperature decrease and cloud cover development. Although the kind of ozone plume regional transport observed for pollution episodes in Paris

are specific to the geographical position of this megacity, the need to take these regional contributions into account in order to understand the variability of pollution episodes in megacities is consistent with what has been observed in past campaigns, e.g. in North America.

# 1  Introduction

Ozone pollution poses significant challenges for air quality management during summer due to its harmful effects on human
health and ecosystems (Fowler et al., 2008). As a secondary pollutant, $O_3$ results from the interaction of sunlight with primary pollutants like nitrogen oxides (NOx) and volatile organic compounds (VOCs), summer months being most conducive to its formation (Monks et al., 2015). These interactions are strongly influenced by atmospheric conditions, which vary within the planetary boundary layer (PBL), the part of the atmosphere where the surface emissions directly affects weather and air quality. Vertical profiling of $O_3$ within the PBL is essential for several reasons. Firstly, the production and destruction of $O_3$
at different altitudes in the PBL depend on the vertical distribution of precursor chemicals and meteorological conditions (Zaveri et al., 2003). Secondly, vertical transport processes within the PBL, such as turbulent mixing and convective uplift, significantly impact the distribution and concentration of $O_3$ and its precursors (Seinfeld and Pandis, 2016). Thirdly the $O_3$ plume from urban centers in the context of megacities, where large volumes of pollutants are emitted, can be transported across urban scales and can influence regional air quality significantly. For instance, the interplay between local emission
sources in large urban areas and regional meteorological patterns can result in the formation of extensive $O_3$ plumes that affect large geographical regions (Couillard et al., 2021; Ma et al., 2021). The summer 2022 ACROSS (Atmospheric ChemistRy Of the Suburban foreSt) measurement campaign as part of the PANAME (PAris region urbaN Atmospheric observations and models for Multidisciplinary rEsearch) project employs advanced techniques like $O_3$ lidar, backscatter microlidar, Doppler lidar, radiosounding and aircraft measurements to characterize the vertical structure of the low troposphere in the Paris city
center area. This approach enables us to dissect the complex interactions between $O_3$, its precursors, and meteorological factors at various altitudes in the PBL. The $O_3$ data gathered provides insights into the mechanisms driving pollution episodes and aids in the identification of primary factors contributing to high $O_3$ events.

Over the past two decades several campaigns have focused on understanding $O_3$ pollution episodes in cities. The "Etude et Simulation de la Qualite de l'air en Île-de-France" ESQUIF) project was conducted in the Paris region (Vautard et al., 2003),
the main focus being the analysis of the contrast between summer and winter conditions in the relative contribution local $O_3$ photochemistry compared to regional transport. The ESQUIF campaign results demonstrated that the Paris area was well suited to study the urban heat island (UHI) effect in pollutant distribution due to enhanced turbulence inside boundary layer (Sarrat et al., 2006). During the ESCOMPTE campaign held in 2001, the focus was on the fate of the Marseille area urban and industrial emissions on $O_3$ formation in the context of very complex meteorological conditions with land-sea breeze and orographic
effects (Drobinski et al., 2007). Ground based UV DIAL $O_3$ lidar and aircraft observations demonstrated the sensitivity of the lowermost tropospheric vertical $O_3$ distribution to mesoscale dynamical processes (Ancellet and Ravetta, 2005). Several campaigns took place in North America to characterize high $O_3$ summer concentrations: ~~observed in Southeastern USA and in~~

California: Texas Air Quality Study (TexAQS) 2000 and 2006 and TRacking Aerosol Convection ExpeRiment - Air Quality (TRACER-AQ) 2021 in Southwestern USA (Daum et al., 2004; Senff et al., 2010; Liu et al., 2023), California Research
at the Nexus of Air Quality and Climate Change (CalNex), California Baseline Ozone Transport Study (CABOTS) 2016, Las Vegas Ozone Study (LVOS) 2016 and 2017 in California (Ryerson et al., 2013; Langford et al., 2022; Faloona et al., 2020), Long Island Sound Tropospheric Ozone Study (LISTOS) 2018 and 2019 in New York City (Couillard et al., 2021). During these campaigns extensive use of aircraft and lidar were conducted to better understand the sources and formation mechanism of $O_3$ plumes (Langford et al., 2019). Results of LISTOS, CABOTS and TRACER-AQ show that meteorology
and boundary layer heights are significant parameters influencing the vertical distribution of $O_3$ in these areas. Sullivan et al. (2017) demonstrated that residual $O_3$ layer reincorporation with mixed layer development contributes to a significant part of surface $O_3$ concentration increase in the afternoon. Contribution of long range transport of $O_3$ has been also analyzed using airborne differential absorption LIDAR (DIAL) and satellite. For example it was shown that regional transport of $O_3$ from Asian emissions over the North Pacific Ocean to California is responsible for a significant part of lower tropospheric $O_3$ increase in
Summer (Lin et al., 2012; Langford et al., 2017) and that stratospheric-tropospheric exchanges (STE), forest fires and Asian pollution significantly control baseline ozone and therefore $O_3$ pollution in urban area in North America (Langford et al., 2022; Wang et al., 2021; Faloona et al., 2020).

In the present paper the focus will be again on the Paris area taking advantage of the ACROSS campaign held in Summer 2022 with numerous aircraft flights around Paris and several remote sensing lidar and radar observations carried out in June and
July. Several $O_3$ pollution episodes have been encountered during this period. The presentation of the $O_3$ vertical observations available during this period as well as a preliminary analysis of the respective contribution of the urban boundary layer structure and of the $O_3$ plume regional transport are the main objectives of this paper. The latter has been extensively discussed for North American campaigns listed hereabove, but it is not clear if similar conclusions can be drawn for the Paris area about the role of elevated ozone concentrations transported from outside the megacity area. The Paris area is also different from the places with
complicated pollution plume recirculation due to orography or land-sea breeze meteorological forcing where many previous campaigns took place in Europe or North America. Therefore it is relevant to present a study specific to the development of ozone pollution episode in the Paris area.

The overall description of the $O_3$ variability during the ACROSS campaign and the selection of the pollution events analyzed in this work are presented in section 3.1. This section focusses on lidar observations and the comparison with aircraft and model
data. Section 4.1 first shows to what extent the $O_3$ measurements discussed in this work are relevant for studying the summer day-to-day variability of ozone in the lower troposphere in Paris, including the potential input from satellite observations. Section 4.2 presents the analysis of the regional $O_3$ transport during ACROSS since this process has been recognized during the past campaigns as a significant source of variability. Sections 4.3 and 4.4 summarize the main characteristics of the summer pollution episodes encountered during ACROSS and put the results into a broader perspective by comparing them with those
of past measurement campaigns.

## 2 Description of observation and modelling tools

### 2.1 In-situ surface observations

Numerous observations are available in the Paris area to monitor hourly-averaged $O_3$ concentration and temperature. We will focus in this work on three monitoring sites located in (i) the Paris 13 station located at 60 m ASL in a park not directly influenced by traffic emissions, (ii) the top of the QUALAIR University Zamansky tower at 125 m ASL (iii) the 3rd floor of the Eiffel tower at 310 m ASL (Fig. 1). Since no temperature are available at the Paris 13 station, the Luxembourg park temperature at 46 m ASL has been used to characterize the surface temperature. The accuracy of the $O_3$ measurements is around 5 µg.m$^{-3}$. $O_3$ concentrations will be given in µg.m$^{-3}$ as this is the true quantity measured by the lidar and ozone observations made by the Air Qulity network are also given in µg.m$^{-3}$. The tower observations have been used to characterize the temporal evolution of the surface layer lapse rate and the $O_3$ vertical gradient near the surface. The latter is very useful to measure the $O_3$ vertical profile down to the ground as the QUALAIR lidar is blind below 250 m AGL.

### 2.2 Ozone vertical profiles

The observations discussed in this work have been carried out during the ACROSS campaign from June 13, 2022 to July 13, 2022. $O_3$ vertical profiles are obtained from a UV DIAL instrument installed on the Sorbonne Université campus. The instrument is described in Klein et al. (2017); Ancellet and Ravetta (1998) and provide observations in the altitude range 0.3 to 5 km during nighttime and up to 2.7 km during daytime. Only daytime measurements have been carried out during ACROSS-2022 as the lidar could not be remotely controlled during this campaign. Although the DIAL sampling rate is 15 s, the $O_3$ vertical profiles are usually hourly-averages to match the surface data time resolution and to improve the lidar signal-to-noise ratio above the planetary boundary layer top. The accuracy of the lidar observations is altitude-dependent being of the order of 7 µg.m$^{-3}$ below 1000 m and occasionally increases up to 20 µg.m$^{-3}$ above 2 km at midday (Klein et al., 2017). The latter is due to elevated background skylight noise at noon or to a reduction in the number of averaged lidar shots during scattered cloud occurrence at altitudes below 2 km. The vertical resolution is less than 100 m at a 250-m altitude and of the order of 500 m at a 2500-m altitude. $O_3$ concentrations will be given in µg.m$^{-3}$ in this paper as it is the true quantity measured by the lidar and ozone observations made by the Air Quality Network are also given in µg.m$^{-3}$ (conversion to mixing ratio at 25°C and 1 atm is 1 ppbv = 1.96µg.m$^{-3}$).

Ozone in-situ measurements on IAGOS (In-service Aircraft for a Global Observing System) aircraft provide a vertical profile of $O_3$ during take-off and landing at the Paris Charles de Gaulle (CDG) airport (Nédélec et al., 2015). Typical aircraft trajectories during landings (early morning flights before 6 UT) and take-offs (midday flights after 10 UT) are shown in the supplementary document (Fig. S1, S2). The horizontal domain, where the aircraft remains at altitudes of less than 3 km, does not exceed a radius of 40 km around CDG airport. The aircraft is never above the city center when it flies below 3km. The aircraft location is generally northeast of Paris between 2.5°and 3°E during takeoff (afternoon flights) except on July 13 when the aircraft position is northwest of Paris. The aircraft positions during landing (early morning flights) are generally within a 20km x 50km box either northwest or northeast of Paris. The accuracy of the IAGOS $O_3$ measurements is better than ±2

ppbv/±2% (Thouret et al., 1998) and the vertical resolution of the $O_3$ profile is of the order of 30 m. The respective positions of the $O_3$ DIAL and of the CDG airport are shown in Fig.1.

## 2.3 PBL height characterization

Two instruments have been used to characterize the PBL evolution nearby the QUALAIR $O_3$ lidar: an autonomuous 808-nm microlidar (SLIM) derived from the IAOOS instrument developed by CIMEL and LATMOS (Pelon et al., 2008; Mariage et al., 2017) and meteorological radiosondes launched 4 times a day for 6 days of pollution in June 2022. The SLIM lidar is routinely operated at Jussieu QUALAIR facility on a 24-hour/7-day basis. It provided observations during the full ACROSS campaign using an automated procedure. In this procedure, the raw backscattered signal is first normalized using the integrated attenuated backscatter signal on water cloud layers (O'Connor et al., 2004). The attenuated backscatter signal is derived from the SLIM lidar signal after calibration and correction of the geometrical factor (Pelon et al., 2008; Mariage et al., 2017). The attenuated backscatter is used to identify clouds on the basis of a lidar signal attenuated backscatter above a predefined threshold (0.25 km-1sr-1). It is then inverted to derive the backscattering coefficient in aerosol regions using a forward inversion procedure (Klett, 1985). A standard lidar ratio value of 40 sr is used corresponding to urban aerosol. Further refinements in the analysis can be performed to derive more accurate aerosol and cloud optical properties, but are not used here. The analysis is performed on one (the acquisition time) and ten minutes files.

The PBL height and the top of the residual boundary layer (RBL or RL, which is remain of the previous PBL development) are derived from the vertical structure of the aerosol backscattering coefficient and its variance as markers of the turbulent activity developed in the unstable summer boundary layer (Stull, 1988). A simple approach based on the analysis of the gradients is used following previous studies (Dupont et al., 1994; Flamant and Pelon, 1996; Menut et al., 1999). A combination of information is used to mix backscattering and variance-derived heights in order to identify PBL and RL heights. It requires that significant vertical motions can be identified, as it is the case during daytime. The signature of such dynamics in the lidar signal is a coincident variance peak and a backscattering gradient (Menut et al., 1999). In the decay phase, or in the nocturnal layer development, the PBL height can be estimated from the variance as linked to residual turbulence activity (Stull, 1988). The RL height can be derived from the backscattering coefficient gradient, as particles are maintained in the atmosphere close to the maximum height (depending on particle size and subsidence) reached by the PBL during the day (or the day before for the morning period).

The meteorological radiosondes have been used to plot the thermodynamic skew-T diagram in order to determine the depth of the layer limited by the adiabatic ascent. It also allows to capture the lifting condensation level (LCL) where cloud base can be expected and level of free convection (LFC) above which fast vertical motion and deep convection can occur. The python library metpy.calc.lcl has been used for the automatic retrieval of LFC and LCL. It is complementary to the SLIM estimate of the PBL vertical structure. The SLIM lidar and the radiosounding site locations are shown in the Fig.1.

## 2.4 Satellite observations: IASI

IASI (Infrared atmospheric sounding interferometer) is a nadir-viewing spectrometer (Clerbaux et al., 2009) that records the thermal infrared emission of the Earth-atmosphere system between 645 and 2760 $cm^{-1}$ from the polar Sun-synchronous orbiting meteorological Metop series of satellites. Metop-A, -B and -C were successively launched in October 2006, September 2012 and November 2018. IASI provides global coverage of the Earth twice a day (at 9:30 and 21:30 mean local solar time) with a set of four simultaneous footprints of 12 km diameter on the ground at nadir. Thanks to IASI high spectral resolution of 0.5 $cm^{-1}$ and a low radiometric noise below 0.4 K, vertical composition of various trace gases such as $O_3$ can be assessed in the troposphere (Eremenko et al., 2008; Boynard et al., 2009; Viatte et al., 2011; Safieddine et al., 2013; Wespes et al., 2018).

In this study, we use the IASI $O_3$ profiles retrieved from the FORLI (Fast Optimal Retrievals on Layers for IASI) algorithm (Hurtmans et al., 2012) that can be downloaded from the AERIS portal (http://iasi. aeris-data.fr/O3/; Aeris, 2024). The FORLI-$O_3$ products (profiles and columns) have undergone a series of validation using available ground-based, aircraft, ozonesonde and other satellite observations over local areas and/or short time periods (Antón et al., 2011; Dufour et al., 2010; Pommier et al., 2012) and more recently at global scale over a 10-years period (Boynard et al., 2016, 2018; Keppens et al., 2018). IASI data and $O_3$sonde measurements are in agreement in the troposphere at mid-latitudes (differences of 11-13%) with a significant vertical sensitivity in the troposphere (Boynard et al., 2018). For this work, IASI/Metop-B and -C pixels located within the ACROSS domain (48.84°N-49°N, 2°E-2.5°E) associated with a fractional cloud coverage of 13% or less and filtered by retrieval quality flags (see Boynard et al. (2018)) have been selected. The $O_3$ 0-3 km partial columns can be retrieved for both morning ($\approx$ 9:30 LT, called AM) and evening ($\approx$21:30 LT, called PM) overpasses.

## 2.5 CAMS ozone plume modelling

Copernicus Atmosphere Monitoring Service (CAMS) provides ENSEMBLE model hourly analysis of $O_3$ and $NO_2$ concentration at 5 levels (500m, 750m, 1000m, 2000m, 3000m) with an horizontal resolution of 10 x 10 km. Up to eleven air quality models are used to build the ENSEMBLE analysis reducing the sensitivity to model error (Marécal et al., 2015; Inness et al., 2019). In this work analysis have been used at 3 daily time steps 6 UT, 12 UT and 18 UT to map the $O_3$ and $NO_2$ plume positions over Northern France. The quality of the tropospheric $O_3$ CAMS daily analysis is generally in good agreement with $O_3$sondes and IAGOS aircraft observations at Northern mid-latitudes especially to simulate the formation of regional $O_3$ plumes during the summer (Wagner et al., 2021).

## 3 The ACROSS ozone vertical profile dataset

### 3.1 Selection of the ozone measurement period

The ACROSS-2022 campaign took place during three interesting periods with $O_3$ concentrations above 100 $\mu g.m^{-3}$ and surface temperature above 30°C. The time evolution of the surface hourly $O_3$ and temperature means are shown in Fig. 2 for the 3 stations located at different altitude levels between 40m and 310m ASL. Twelve days corresponding to the red arrows

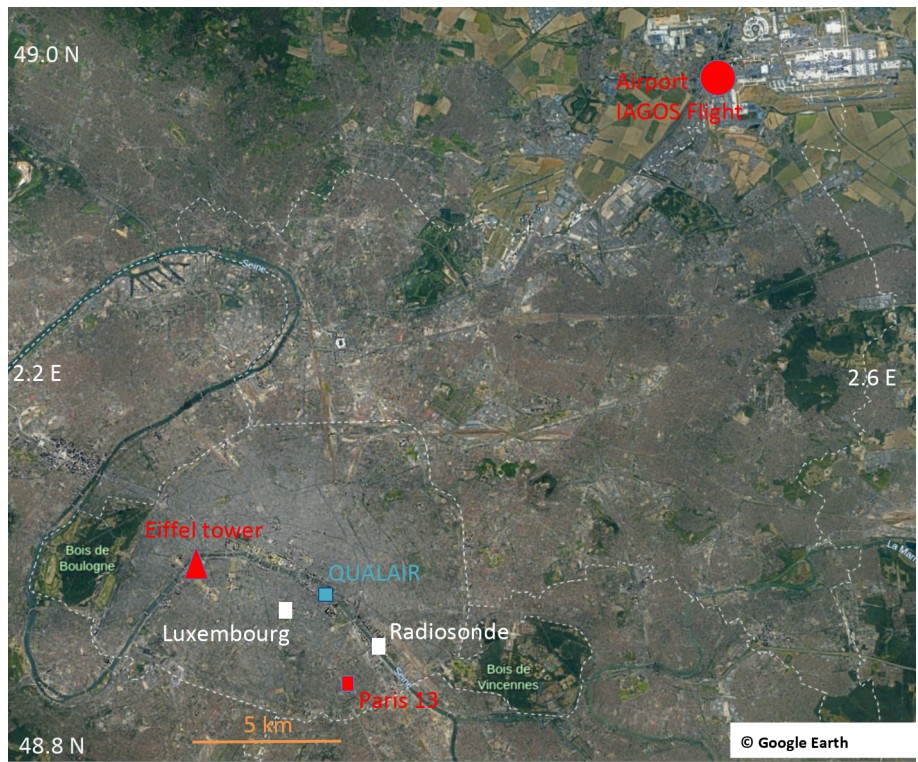

**Figure 1.** Google Earth map of the Paris area with $O_3$ and meteorological measurement positions during the ACROSS-2022 experiment. The blue mark is the position of the QUALAIR station including the $O_3$ DIAL (33m ASL) and instruments operated on the top of the University Zamansky tower (125 m ASL). The red triangle is the Eiffel tower 3rd platform (310 m ASL), the red square is the AIRPARIF PARIS 13 in-situ $O_3$ monitor in Parc de Choisy (50 m ASL), and the red circle is the IAGOS aircraft take-off and landing airport. White squares show the Parc du Luxembourg surface meteorological station (46 m) and the meteorological radiosounding station.

in Fig. 2 have been chosen to determine how the characterization of vertical $O_3$ profiles and the spatial distribution of the pollution plume on a regional scale can be used to better intercompare these different $O_3$ pollution episodes. Unfortunately no lidar data are available after July 14, e.g. during the $O_3$ pollution episode on July 18. This is why this last pollution event is not considered in this work. The 500m-CAMS $O_3$ distribution is a good proxy to track the day to day spatial distribution of the $O_3$ plume at the regional scale, this plume being related to both the regional emissions of Western Europe and the urban emissions from the Paris area. They are shown at 18 UT when $O_3$ concentrations reach their daily maximum in Fig.3 and 4 for the 12 days identified in Fig.2. The first period with elevated $O_3$ concentrations took place from June 14 to June 18. This period was characterized by the highest $O_3$ concentrations (170 µg.m$^{-3}$) recorded within the city center, but also by $O_3$ concentrations $>$140 µg.m$^{-3}$ over a large fraction of Northern France according to the CAMS simulations (Fig. 3). The second time period from June 21 to June 28 is rather typical of summer sunny days with ground temperatures near 30ºC and moderate $O_3$ pollution of the order 110 µg.m$^{-3}$ on June 21, 22 and 28. The CAMS simulations show a well defined $O_3$ plume west of Paris on June

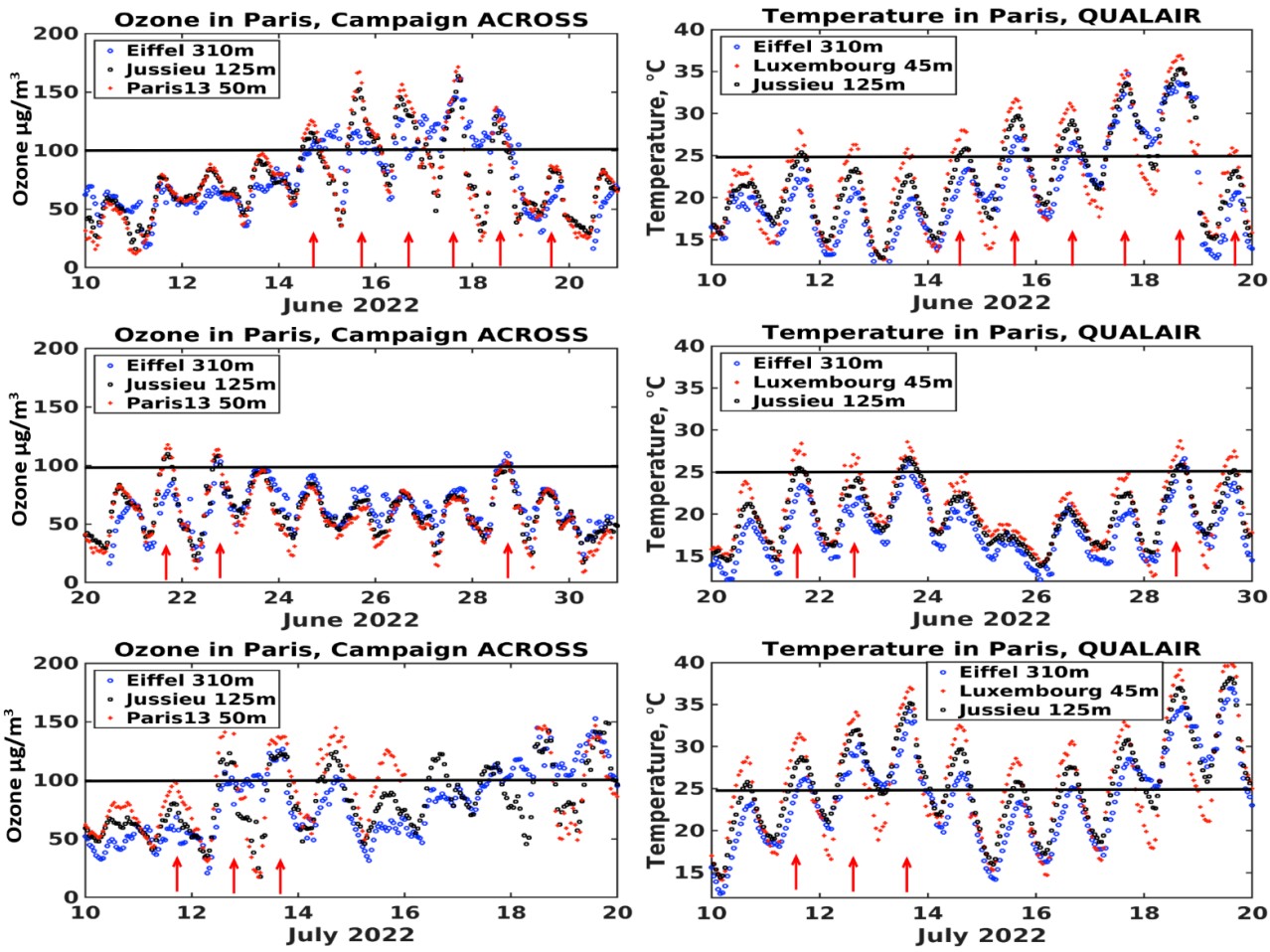

**Figure 2.** Surface O₃ concentration and temperature 10-day time evolution of the hourly mean during the ACROSS-2022 campaign in blue for the Eiffel tower top (310 m ASL), in black for the University Zamansky tower top (125 m ASL) and in red for the Paris 13 O₃ sensor and for the Luxembourg park meteorological station (50 m). Days selected for the analysis of O₃ pollution events are shown by the red arrows.

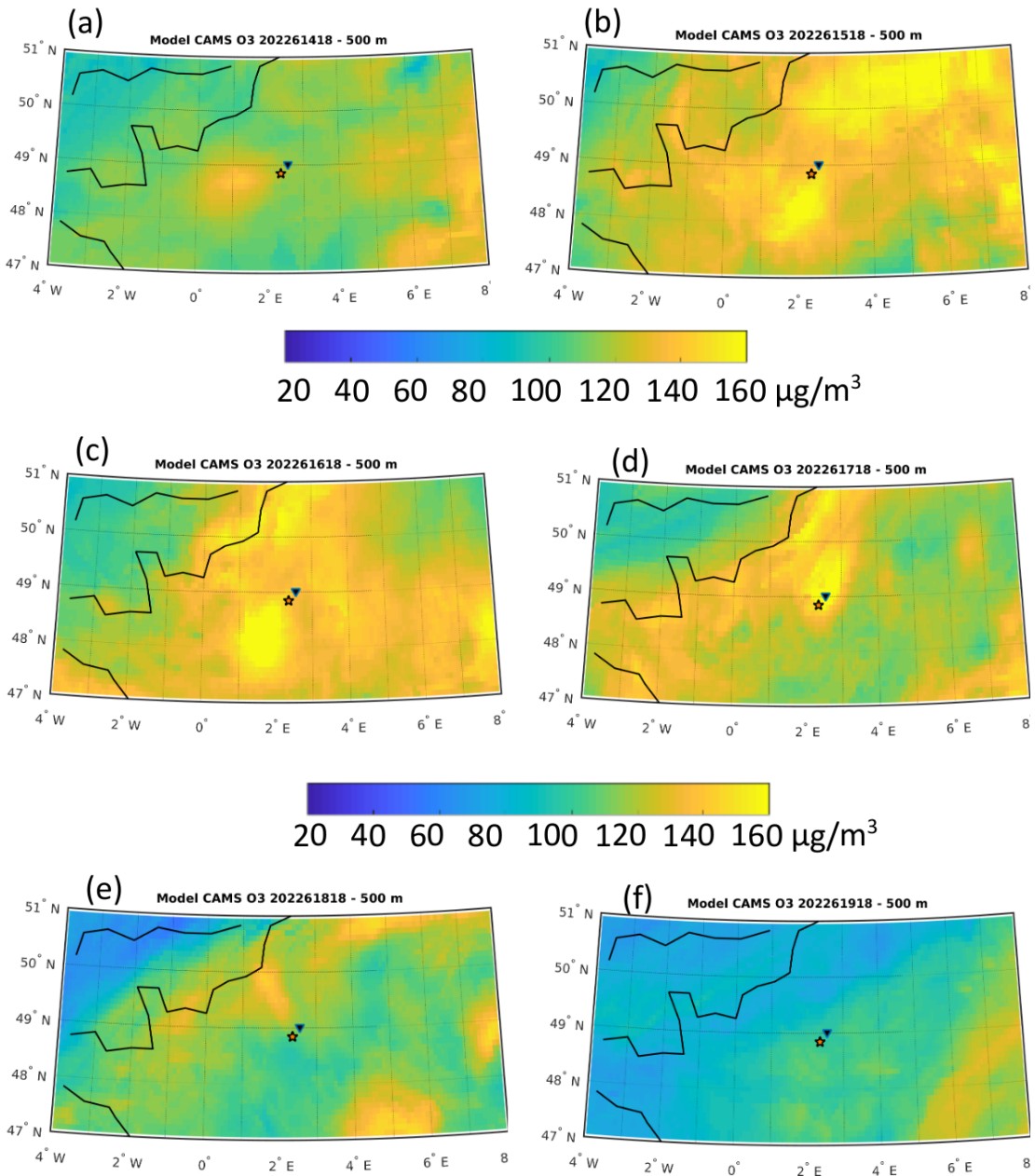

**Figure 3.** Ozone concentration distribution of the CAMS ensemble mean at 500 m above Northern France from June 14 (a) to June 19 (f), 2022 at 18 UT. The orange star and dark-blue triangle are respectively the DIAL position and the CDG airport. The color scale represents the $O_3$ concentration in $\mu g.m^{-3}$.

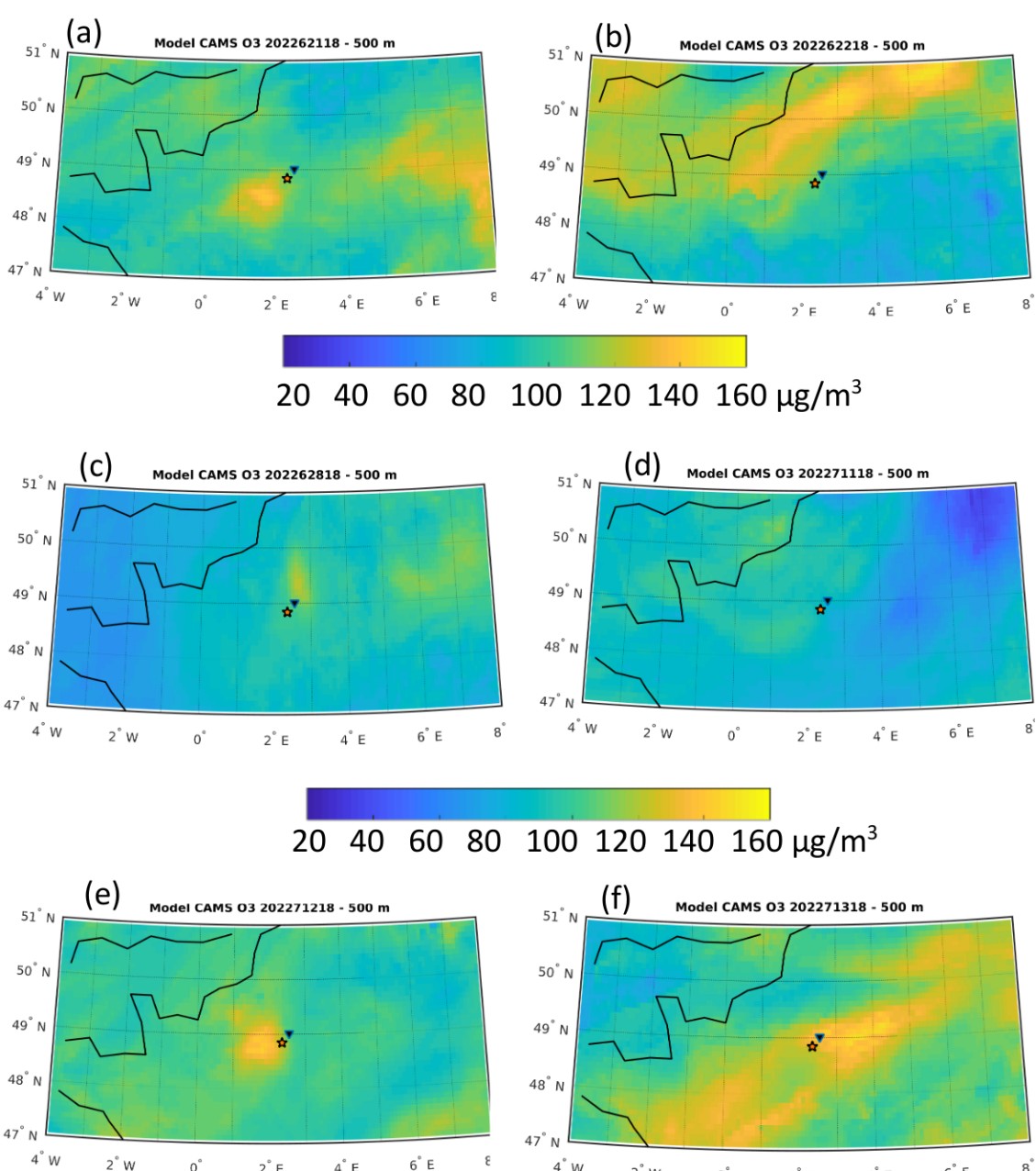

**Figure 4.** As Fig.3 for June 21 (a) to June 22 (b), June 28 (c) and July 11 (d) to 13 (f), 2022.

21 and June 22 (Fig.4a,b) with $O_3$ concentrations approaching $150\,\mu g.m^{-3}$. June 28 is also interesting as the Paris city $O_3$ concentrations below 300m are similar to the June 21/22 episode, while the June 28 CAMS $O_3$ concentrations in the plume located North of Paris (Fig.4c) remain below $130\,\mu g.m^{-3}$. The third $O_3$ pollution period took place on July 11 until July 13 with regional $O_3$ plumes (Fig.4d,e,f) somewhat similar to the June 21/22 case study. The city center $O_3$ concentrations are however as high as $140\,\mu g.m^{-3}$ approaching the values encountered during the June 14/18 episode. Both the $O_3$ and temperature vertical gradient between the surface and 300 m at the time of the daily maximum decrease on the last days of each pollution episode. Measuring the $O_3$ vertical profiles by the UV DIAL and IAGOS aircraft above the top of the Eiffel Tower is mandatory to understand to what extent the structure and intensity of the $O_3$ plume described by the CAMS simulations can explain the surface measurements in central Paris. A better understanding of the weakening of the surface $O_3$ vertical gradient between 0 and 300 m altitude will benefit also from the lidar and aircraft observations at altitudes above 300 m.

### 3.2  Boundary layer diurnal variation

An $O_3$ layer is generally observed during the morning hours above the nocturnal surface layer in the RL (Neu et al., 1994; Klein et al., 2019). It is generally an $O_3$ reservoir with limited NOx titration and $O_3$ deposition. This $O_3$-rich air in the RL can be mixed down into the surface boundary layer effectively (Caputi et al., 2019). The analysis of the boundary layer height diurnal variation using the microlidar SLIM is necessary to characterize the PBL growth during the day and the downward mixing of the RL. The diurnal variations of the 10-min aerosol backscatter vertical profiles measured by SLIM are shown in Fig.3 to S6 for the 12 days with elevated $O_3$ concentrations. The PBL height (PBLH) and RL height (RLH) are derived using the methodology described in section 2.3 and are shown using respectively blue star and blue circle in the supplementary document (Fig.S3 to S4). These plots are also useful to identify the occurrence of long range transport of aerosol plumes in the free troposphere above the PBL, e.g. the Saharan dust plumes observed in the 2-4km altitude range from June 15 to June 18 (Fig.S3b, Fig.S4a,b) or the recirculation of the European continental aerosol (Fig.S5a,b). The aerosol plume attribution was based on linear 532 nm depolarization ratio larger than 0.2 measured by the CIMEL lidar measurements of the QUALAIR station and based on long range transport modeling in section 4.2. Another interesting feature is the overall difference of the aerosol backscatter magnitude within the PBL when looking at the first (June 15-18) and at the last (July 12-13) heat wave episode. The latter with no dust plume aloft corresponds to less aerosol backscatter within the PBL. Large 808 nm aerosol backscatter values above $0.01\ km^{-1}sr^{-1}$ (yellow pixels in Fig.S3 to S6 correspond to cloud layer formation at the top and above the PBL. The 1-min high resolution cloud observations of the SLIM lidar have been also used to filter out the cloudy DIAL lidar observations when retrieving the $O_3$ profile.

Data of the 16 meteorological radiosondes are also shown in Table 1 and in Fig.S3 to S5 using thermodynamic skew-T diagrams. The bottom altitudes of inversion layers (ILH) detected by the radiosondes below 4 km shown in Table 1 are retrieved using layers with potential temperature vertical gradient larger than 15 K/km. They can be compared with the SLIM PBLH and RLH. There is a good agreement between both the lidar retrieval and the analysis of the meteorological radiosondes especially for the timing of the PBL growth and the low thickness of the surface layer around 00 UT. The PBLHs generally remain below 2 km between June 14 and 22 (Fig.S3 to S5b), except on June 18 with a fast rising of the PBLH in the evening (Fig.S4b). The

**Table 1.** Comparison of the 808-nm microlidar SLIM Planetary Boundary Layer (PBL) and Residual Layer (RL) heights with the meteorological radiosounding inversion layer (IL) bottom altitudes observed below 4 km

| Date | 06/14 | 06/16 | | 06/17 | | | 06/18 | | | 06/19 | | 06/22 | | 06/28 | | |
|---|---|---|---|---|---|---|---|---|---|---|---|---|---|---|---|---|
| Hour,UT | 00 | 12 | 20 | 00 | 12 | 20 | 00 | 12 | 20 | 00 | 12 | 16 | 20 | 00 | 12 | 20 |
| First IL height,km | 0.2 | 1.4 | 1.6 | 0.2 | 1.2 | 0.25* | 0.2 | 1.0 | 0.3* | 0.5 | 0.5* | 0.8 | 1.0 | 0.4 | 2.1 | 2.3 |
| Second IL height,km | 1.4 | | 2.2 | | | 1.2 | | 3.5 | 3.5 | 3.5 | 1.5* | 2.0 | 3.8 | 2.4 | | |
| Lidar PBL height,km | 0.3 | 1.3 | 1.0 | 0.25 | 1.2 | 0.5 | 0.25 | 1.0 | 3.5 | 0.25 | 0.7 | 0.9 | 1.0 | 0.4 | 2.1 | 1.6 |
| Lidar RL height,km | 1.5 | | 1.7 | 1.2 | | 1.1 | 1.02 | | | 3.5 | | | 1.9 | 2.2 | | 2.3 |

*stable layer with thickness $< 50$ m

PBLHs however exceed the 2-km altitude level on June 28 (Fig.S5c) and during the third pollution episode (Fig.S6). It is likely related to a change in the atmospheric circulation due to a change in synoptic weather pattern with anticyclonic downward advection before June 22 and upward advection of marine air from the Atlantic ocean or the North sea on June 28 or on July 11 to 13 (see section 4.2). The largest PBLHs beyond 3 km have been observed on June 18 and July 13 for the highest surface temperatures above 35°C (Fig.2).

The daily maximum of the PBLH generally occurs around 17 UT, while PBLH and RLH decrease below 2 km at 23 UT despite high surface temperatures. Another interesting feature for the downward transport of $O_3$ the following day is the occurrence of RL heights below 1.5 km at 21 UT, followed by a continuous decrease in RLHs after 21 UT. There are 4 days with such behavior: June 14, 16, 22 and July 12. These 4 days in fact correspond to high nighttime surface $O_3$ concentrations above 100 $\mu g.m^{-3}$ (Fig.2), consistent with an efficient downward mixing of RLs in the 0-300m surface layer during the night.

### 3.3 DIAL ozone diurnal variation

The $O_3$ vertical profile are taken from the $O_3$ DIAL observations for the days selected in section 3.1. The time/altitude daytime evolution of the $O_3$ concentration is shown in the left-hand columns of Fig.5 and 6. Data from the surface stations shown in Fig.2 are also included to these figures using the same color-coded scale. They correspond to the pixels with the black cross in Fig.5 and 6. The diurnal cycle observed by the Eiffel and Zamansky tower stations at 125 m and 310 m are consistent with the lidar observation at 300 ASL, following the previous study of Klein et al. (2017). CAMS vertical $O_3$ profiles are also retrieved using the ensemble model data at 5 vertical levels and within the box [48.84°N-49°N, 2°E-2.5°E]. The latter corresponds to an horizontal domain of 36km x 17km including the QUALAIR station and the CDG Airport. The CAMS vertical profiles are shown at 6 UT, 12 UT and 18 UT in Fig.7 when DIAL lidar observations are available. The averages of the morning and mid-

day IAGOS O$_3$ vertical profile up to 3 km are also shown in Fig.7. The averages of the morning and mid-day DIAL O$_3$ profiles are also shown for the time periods of the IAGOS flights in Fig.7. Such a comparison of the three O$_3$ profiles is useful to check if the O$_3$ layers observed by the DIAL in the Paris city center is also present at the scale of the entire Paris Ile de France region and if advection of the regional O$_3$ plume plays a significant role in the O$_3$ diurnal variation in the city center. The PBLH and RLH diurnal variation derived from results of section 3.2 are also included in the DIAL O$_3$ time-altitude cross-sections (Fig.5

and 6) to take into account the role of RL in the O$_3$ vertical profile diurnal variation, but also the possible downward mixing within the PBL of O$_3$-rich or O$_3$-poor layers advected in the free troposphere above Western Europe.

The well-known early morning O$_3$ depletion due to the nighttime O$_3$ deposition and NO$_2$ titration (Güsten et al., 1998) is observed up to 750 m by the DIAL with concentrations as low as 40 µg.m$^{-3}$ before 9 UT. The daily maximum O$_3$ concentrations at the surface and the PBL top is always found after 14 UT when O$_3$ precursor gases are transported upward within the

PBL. The largest daily O$_3$ concentrations (up to 175 µg.m$^{-3}$) in the 500m-1000m altitude layer observed by the DIAL on June 15 to June 17 correspond very well with the three days when elevated CAMS O$_3$ concentrations larger than 140 µg.m$^{-3}$ are present over a large part of Northern France according to Fig.3. The IAGOS and CAMS vertical profiles in Fig.7) show also the largest O$_3$ concentration are observed below 1.5 km over the Paris Ile de France area (150-160 µg.m$^{-3}$) on June 16. The two days with the lowest UV DIAL O$_3$ concentration (below 100 µg.m$^{-3}$) on June 28 and July 11 corresponds to a large fraction

of Northern France with O$_3$ daily maximum at 18 UT below 80 µg.m$^{-3}$ (Fig.4c,d). The IAGOS and CAMS O$_3$ vertical profiles also show concentrations below 100 µg.m$^{-3}$ for these two days (Fig.7).

The depth of the afternoon O$_3$ layer is generally below 1.5 km and corresponds quite well with PBLH (blue star in Fig.5, 6), except on June 19 and June 22 when the O$_3$ layer extend up to 1.5 km while PBLH maximum remains below 1 km. On June 22, PBLH might be underestimated by SLIM since on one hand, PBLH retrieved in section 3.2 rises up 2 km only after

17 UT despite the presence of an aerosol layer up to 2 km at 15 UT and on the other hand, the 16 UT meteorological sounding identifies a well defined ILH at 2 km. The O$_3$ DIAL might help to clear up ambiguity about the PBLH value more in line with the 16 UT radiosounding. On June 19, there is no reason to question the low PBLH of the microlidar SLIM, while advection of the continental pollution plume above the PBL might very well explain the presence of the 130 µg.m$^{-3}$ O$_3$ layer between 1 and 1.5 km (see next section 4.2). There are only three days with both PBLH and DIAL O$_3$ layer depth above 2 km: June 28

and July 12, 13. The IAGOS O$_3$ concentrations also reach 100 µg.m$^{-3}$above 2.0 km on these three days (Fig.7).

The positions of the RLH (blue circle in Fig.5 and 6) are also in very good agreement with O$_3$ concentrations above 100 µg.m$^{-3}$ early in the morning between 0.6 and 1.3 km, when high ozone > 120 µg.m$^{-3}$ are observed in the PBL on the previous day. Regarding the layers at altitude levels above the PBLH or the RLH, there are two periods with large differences in O$_3$ concentrations ($\approx \pm 60$ µg.m$^{-3}$) measured in the free troposphere and in the PBL/RL. First low O$_3$ concentrations less than

80 µg.m$^{-3}$ are observed by the DIAL above 1.5 km on June 17 and 18 (Fig.5b, d) corresponding to the dust plume advection discussed in section 3.2. The IAGOS and CAMS vertical profiles above 2 km (Fig.7) show also O$_3$ concentrations less than 80 µg.m$^{-3}$. Second the June 22 O$_3$ layer in the 1km-2.5km altitude layer (Fig.6c) is different from the other days with free tropospheric O$_3$ concentrations up to 130 µg.m$^{-3}$, while this layer doesn't mix very well with the surface layer during the day. Such a layer with concentration larger than 100 µg.m$^{-3}$ is not present in the CAMS vertical profile (Fig.7) above the Paris

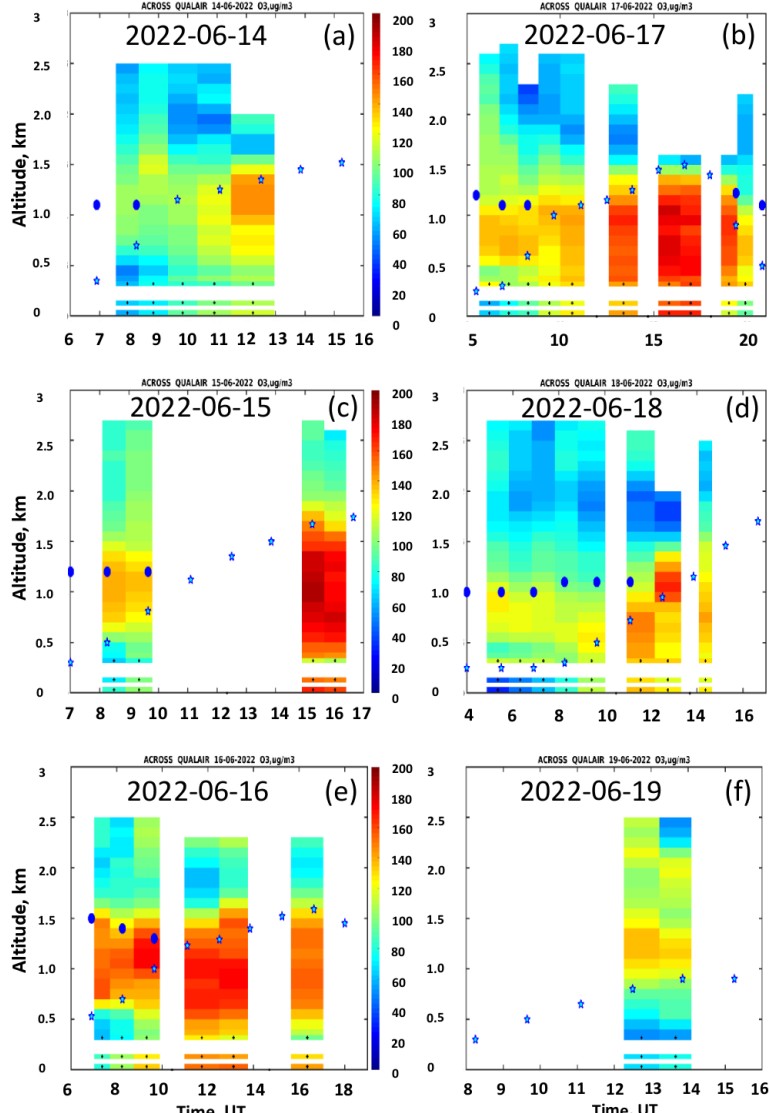

**Figure 5.** Time-altitude $O_3$ concentration daytime evolution in µg.m$^{-3}$ from June 14 to 19, 2022 using DIAL above 300m and in-situ $O_3$ monitor below 300 m (pixel with black cross). The blue star and circle are the SLIM lidar PBLH and RLH show in the supplementary document.

area. The daily average IAGOS profile on June 21 and 22 also exhibits $O_3$ concentrations $>$ 100 µg.m$^{-3}$ at 2 km (Fig.7), but less than the daily average of the city center DIAL $O_3$ observations at 2 km on June 22. This will be discussed in the next section when the regional transport of the air masses transported over Paris is described.

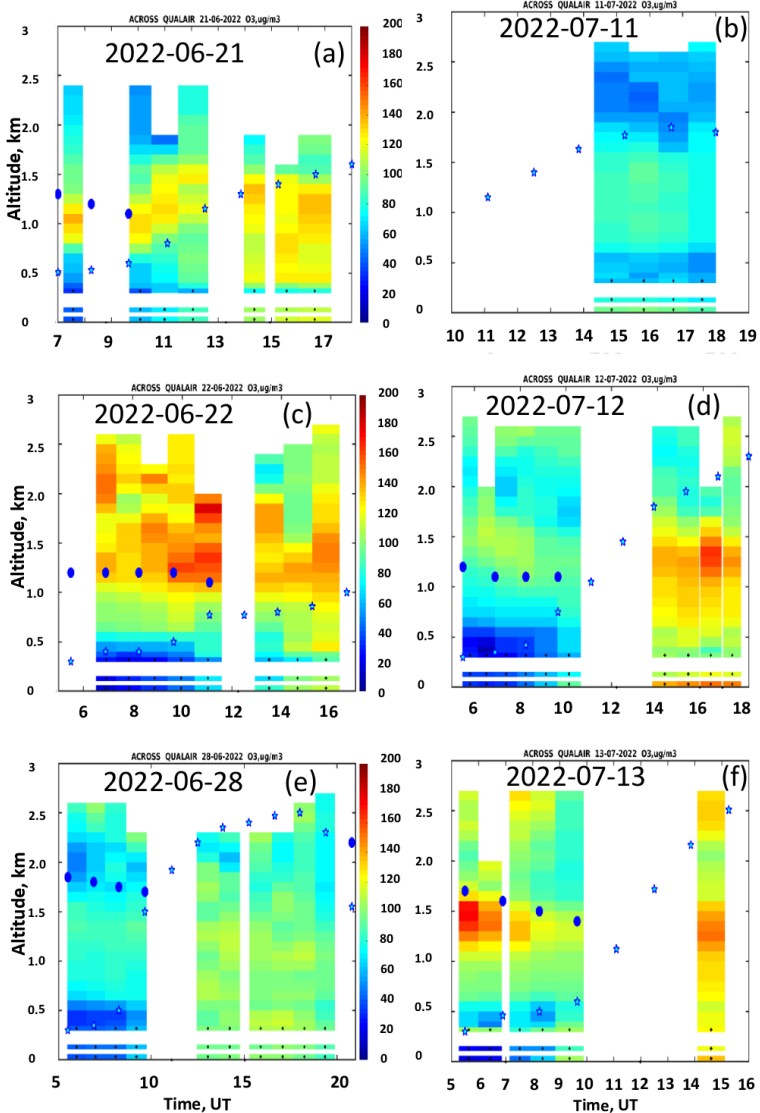

**Figure 6.** As Fig.5 for O$_3$ concentration daytime evolution in µg.m$^{-3}$ for June 21, 22, 28, 2022 and from July 11 to 13, 2022.

## 4  Analysis of the day-to-day variability

### 4.1  O$_3$ lowermost tropospheric column: IASI, CAMS and ACROSS vertical profiles

The IASI O$_3$ 0-3 km partial columns are computed for the period 13 June – 13 July 2022 for both AM and PM overpasses. Comparisons between IASI O$_3$ 0-3 km partial column, ACROSS observations (IAGOS and DIAL) and CAMS simulations are shown in Fig.8. All measurement days, whether or not they corresponded to pollution episodes, are considered in order to

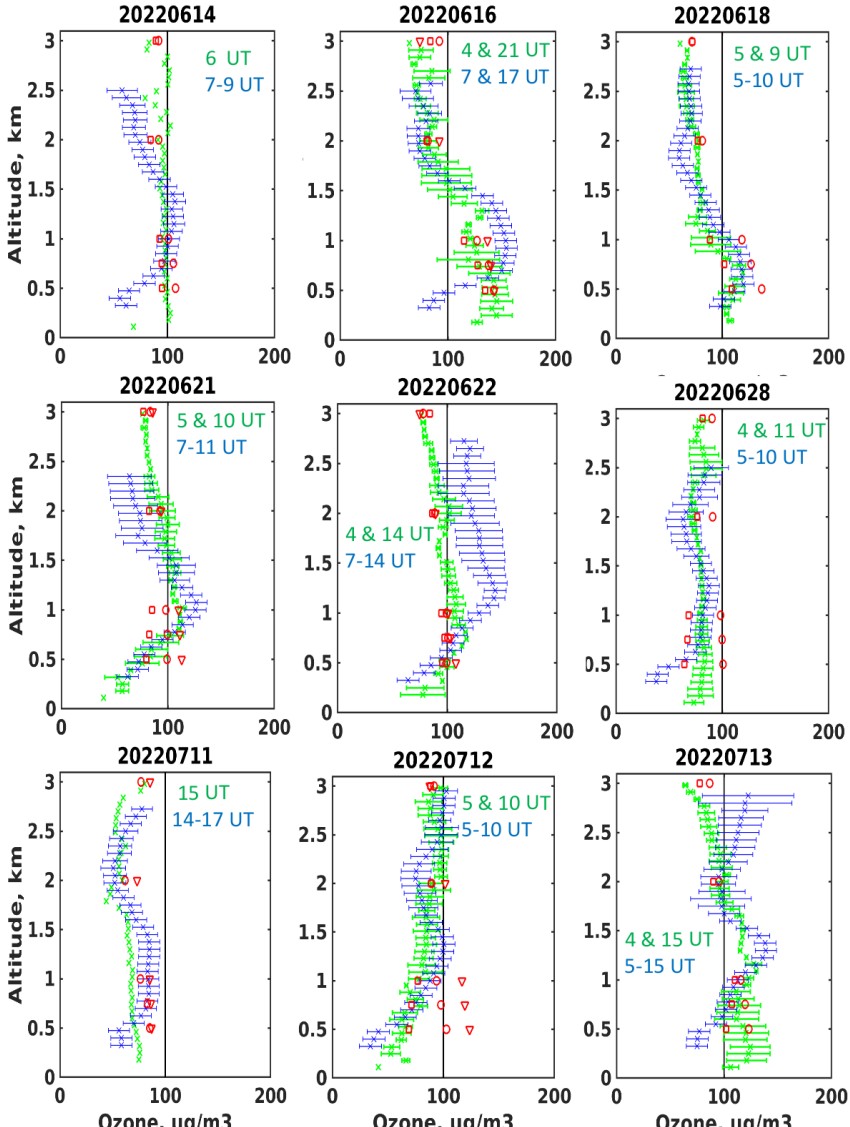

**Figure 7.** Daily mean $O_3$ vertical profiles in $\mu g.m^{-3}$ for the IAGOS aircraft (green) and the corresponding DIAL observations (blue) shown in Fig.5 to 6. Green times in UTC labeled within the figures are the IAGOS measurement times above Paris (two profiles per day except on June 14 and July 11). Blue times below the IAGOS flight times show the selection of the DIAL observations. CAMS model vertical profiles are also shown using horizontal averages of the model concentrations included in the Fig.1 area. CAMS profiles are shown at 6 UT (red □), 12 UT (red ○) and 18 UT (red ▽).

assess $O_3$ variability in the lowermost column during the ACROSS campaign from June 13 to July 14. All the hourly means of the DIAL and IAGOS observations made between 6 UT and 20 UT have been considered as well as the IASI morning and

evening observations. The daily mean of the columns derived from the CAMS ensemble simulations are shown by the red solid lines in Fig.8.

    For the comparison of IASI data against IAGOS/DIAL/CAMS data in the ACROSS domain, a temporal coincidence criterium of $\pm 6$ h is used. For a proper comparison, the IASI averaging kernels (AKs) are applied to the IAGOS, DIAL and CAMS vertical profiles in order to account for the differences in vertical resolution and to remove the dependency of the com-

parison on the a-priori $O_3$ profile information used in the retrieval (Rodgers and Connor, 2003). The IAGOS, DIAL and CAMS profiles ranging from the surface to 3 km are first interpolated on the IASI vertical grid (which corresponds to 0.5, 1.5 and 2.5 km levels) and then degraded to the IASI vertical resolution by applying the IASI AKs and a priori $O_3$ profile according to Rodgers (2000).

$$x_s = x_a + A(x_{raw} - x_a) \qquad\qquad (1)$$

where $x_s$ is the smoothed IAGOS/DIAL/CAMS profile, $x_{raw}$ is the IAGOS/DIAL/CAMS profile interpolated on the IASI vertical grid, $x_a$ is the IASI a priori profile and A is the IASI AK matrix. Incomplete IAGOS/DIAL/CAMS profiles above 3 km are filled with the a priori profile. Based on these criteria, 28, 42, and 19 pairs of observations are found between IASI and the smoothed IAGOS, DIAL, and CAMS data (Table 2), respectively. Over the 13 June – 13 July 2022 period, the averaged IASI column of $7.00 \pm 1.40$ DU is in agreement with the smoothed IAGOS, DIAL and CAMS datasets, with averaged columns of

$8.53 \pm 0.40$, $8.55 \pm 0.49$, and $7.83 \pm 0.12$ DU, respectively.

    Figure 8 shows that the $O_3$ 0-3 km partial columns and variabilities derived from IAGOS, DIAL and CAMS smoothed data are systematically lower than those calculated without taking into account the IASI averaging kernels. Smoothing with the IASI AKs reduces ozone columns and variability because part of the signal information comes from the a priori profile which is constant over time. However, IASI observations exhibit a variability of $\approx 5$ DU (mean of $7.00 \pm 1.40$) over Paris during the

ACROSS campaign, demonstrating that atmospheric signal is present in the retrievals information content with an averaged degree of freedom for signal (DOFS) of 0.22 and 0.08 for morning and evening measurements, respectively. IASI $O_3$ columns are overall lower than IAGOS, DIAL and CAMS raw and smoothed columns, with biases of the order of 1-3 DU, in particular when ozone partial columns above 2 km are low, such as between June 14th and 19th, and between June 29th and July 5th. Inversely, IASI and the smoothed IAGOS/DIAL $O_3$ columns are similar in the case of a high PBL ($> 2.5$ km) or in the case of

high ozone above 2km ($> 100$ $\mu g.m^{-3}$), which are the cases on June 22th, June 28th, and July 12th.

    DIAL measurements show that the diurnal variability of $O_3$ 0-3 km partial column reaches 5 DU during the ACROSS campaign (blue dots and squares in Fig.8), confirming the importance of monitoring $O_3$ profiles at high temporal resolution throughout the day. The day-to-day variability of AM IASI columns is of the same order of magnitude as that of $O_3$ IAGOS, DIAL and CAMS raw $O_3$ columns (5 DU).

**Table 2.** Mean and standard deviation of $O_3$ partial columns (0-3km) in Dobson Unit (DU) derived from IAGOS, DIAL, CAMS, and IASI dataset during the ACROSS campaign between June 13 to July 13 2022.

| Dataset | Raw | Number of observations | Smoothed | Number of observations |
|---------|-----|------------------------|----------|------------------------|
| IAGOS | 11.56±1.93 | 49 | 8.53±0.40 | 28 |
| DIAL | 12.88±2.38 | 52 | 8.55±0.49 | 42 |
| CAMS | 12.00±1.77 | 32 | 7.83±0.12 | 19 |
| IASI AM | 7.75±1.37 | 19 | | |
| IASI PM | 6.25±0.98 | 19 | | |
| IASI | 7.00±1.40 | 38 | | |

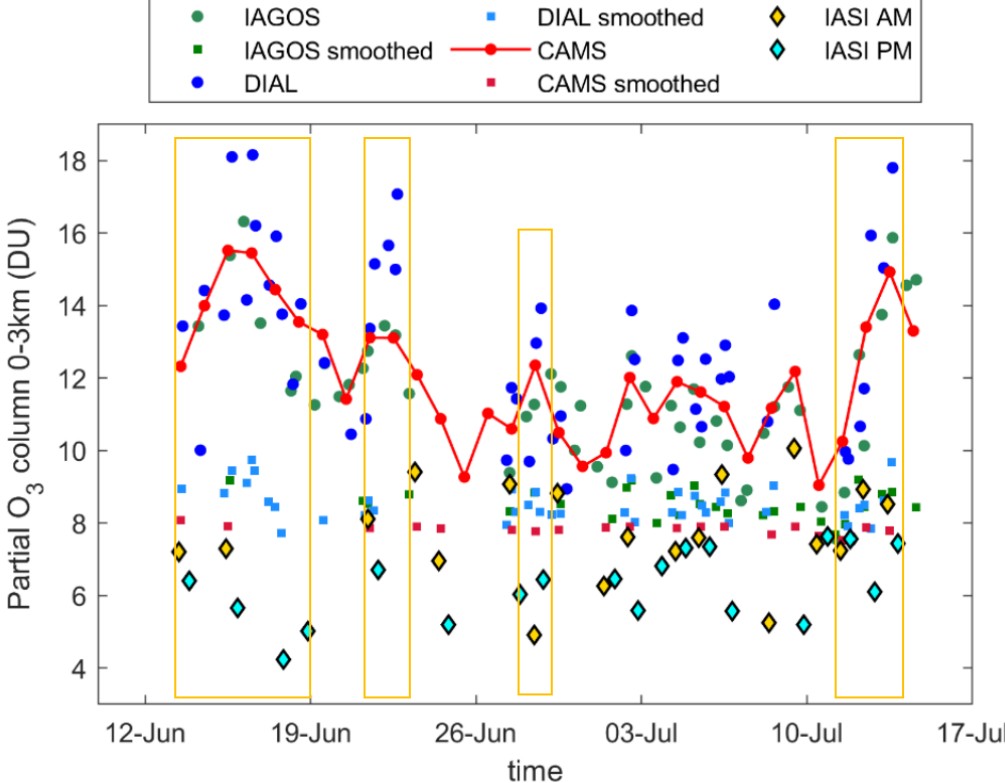

**Figure 8.** Comparison of the 0-3km $O_3$ tropospheric column derived from the ACROSS observations (DIAL in blue and IAGOS in green), CAMS data (in red), and IASI satellite observations (morning – yellow diamonds, and evening – cyan diamonds) calculated in the [48.84°N-49°N, 2°E-2.5°E] box between June 13 to July 13 2022. Circles and squares correspond to the 0-3km $O_3$ partial column and smoothed partial columns degraded to the IASI vertical resolution, respectively. The orange boxes show the pollution days discussed in section 3.

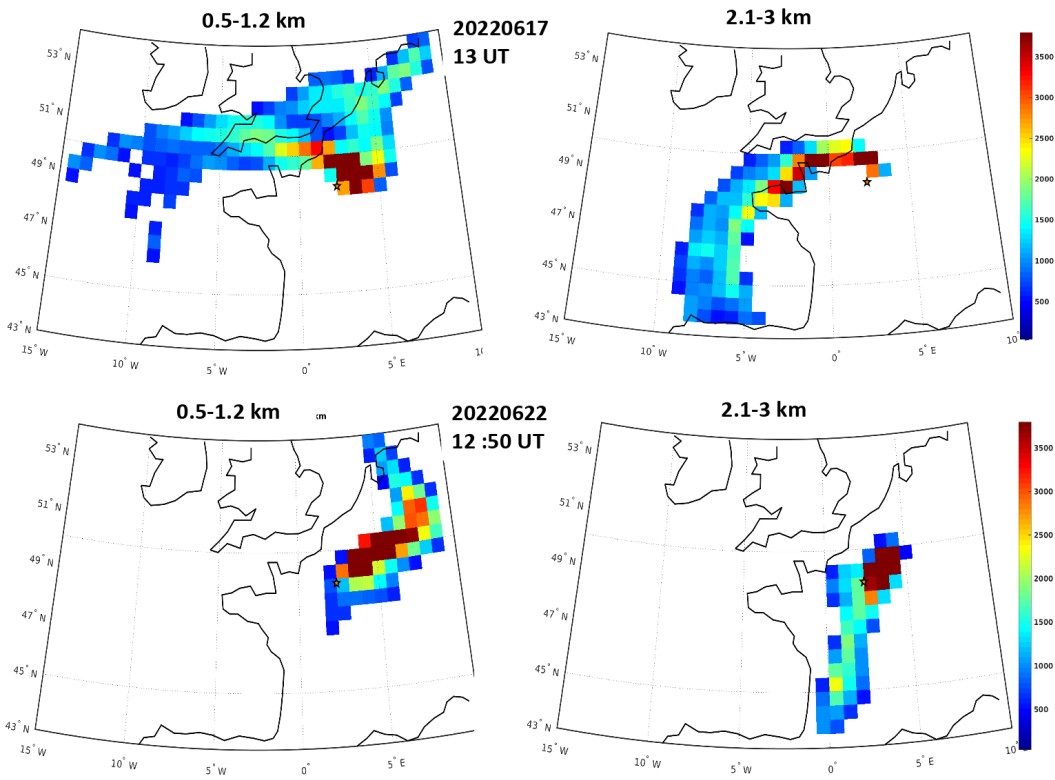

**Figure 9.** Maps of the vertically integrated FLEXPART backward potential emission sensitivity (PES) for particles release at midday in the 0.5-1.2 km altitude range (left column) and in the 2.1-3km altitude range (right column) above the DIAL in Paris city center on June 17 (top) and 22 (bottom). The PES color scale is in s. Integration time is 3 days. The orange star is the DIAL position.

## 4.2 Regional transport of ~~the ozone layer~~

The potential emission sensitivity (PES) of a passive air tracer are calculated with the FLEXPART model version 9.02 initialized with the 1°x1° ECMWF operational meteorological analysis (Stohl and Seibert, 1998; Stohl et al., 2002). The FLEXPART model is run backward over 72 hours with 17000 particles released in boxes 35 km by 35 km wide at different altitude ranges above the DIAL: 0-500m, 0.5-1.2km, 1.2-2.1km, 2.1-3.0km. The 0.5-1.2km and 2.1-3.0km PES maps are shown in the supplementary document (Fig.S7 to S10) using a color scale in s for the vertically integrated residence time of the released particles. One example of the PES maps is shown in Fig.9. All the grid cell altitudes below 3km are cumulated to calculate the mean PES in the lowermost troposphere.The $NO_2$ plume CAMS simulations at the 1 km altitude have been also examined in addition to the $O_3$ CAMS analysis in order to identify the positions of the Paris city plume and the spatial extent of the $O_3$ pollution plume produced at a wider scale (Fig.10). The 1°x1° horizontal resolution of the ECMWF wind analysis is obviously limited for fine tracking of the city plume, but the PES FLEXPART distributions remain very accurate to check to what extent long range transport must be taken into account in the analysis of the city plume. Looking at, first the midday PES distributions of

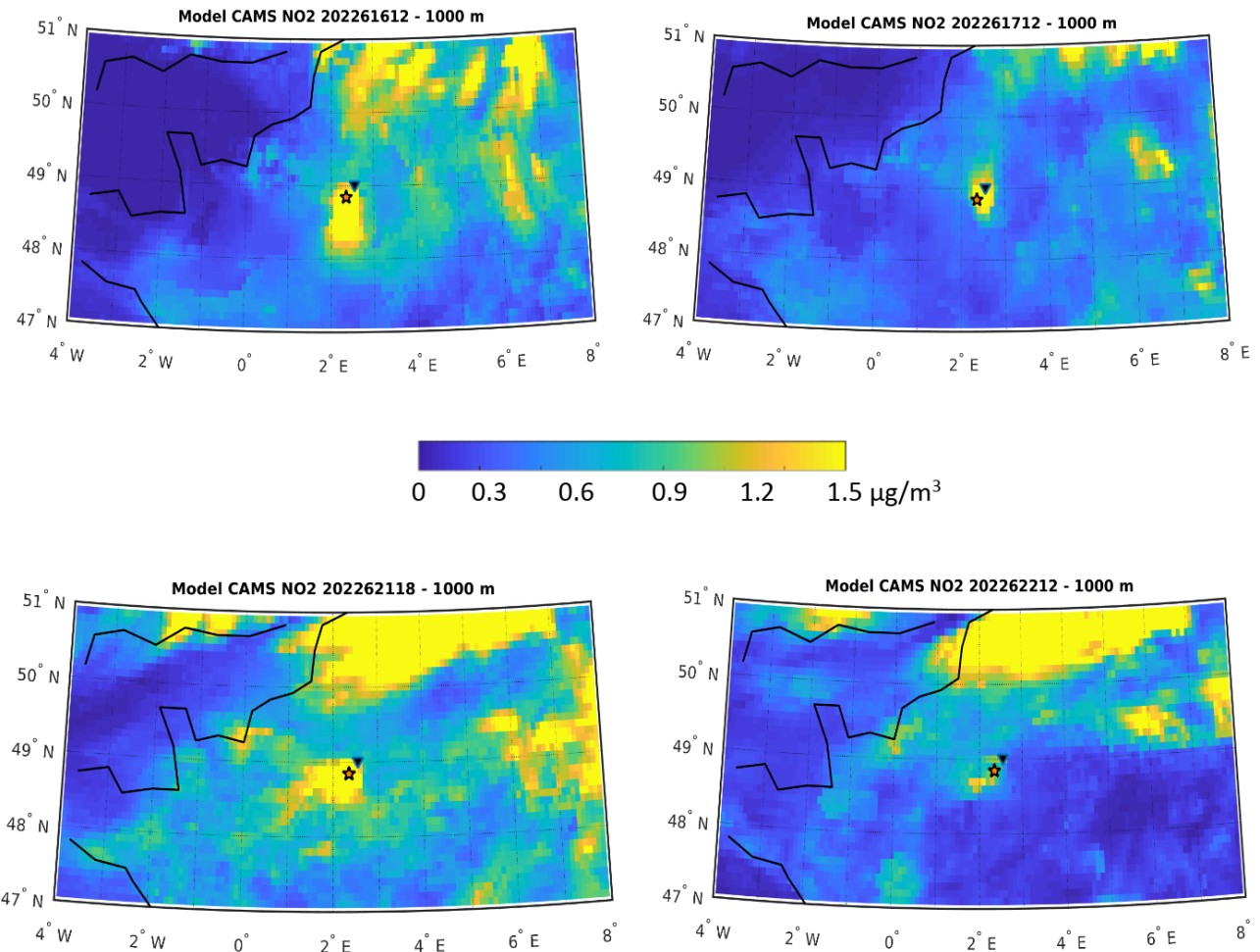

**Figure 10.** CAMS ensemble mean $NO_2$ at 1000 m above Northern France on June 16 and 17 (top row) when dust plume advection at the PBL top is observed by the aerosol lidar and on June 21 and 22 (bottom row) when continental aerosol and $O_3$ plume advection at the PBL top is observed by both lidar. The orange star and dark-blue triangle are respectively the DIAL position and the CDG airport. The color scale is $NO_2$ concentration in $\mu g.m^{-3}$.

particles released either from the PBL or from the free troposphere and, second the CAMS simulations, one can distinguish five horizontal advection patterns:

– June 14, 15: Advection of continental air masses from Benelux and Germany below 1.2 km transport polluted air over Paris since the CAMS simulations (Fig.3) show high $O_3$ plumes over these regions. The anticyclonic circulation below 2 km is also consistent with low PBLH observed in Paris during this time period. Moderate $O_3$ concentrations in the free troposphere are also consistent with a completely different circulation pattern above 2 km bringing cleaner air from the Atlantic ocean and the English Channel.


- June 16 to 18: In addition to the remaining anticyclonic conditions in the lowermost troposphere, long range transport of Saharan dust across Spain and the Atlantic coast above 2 km (Fig.9 and Fig. S7 and S8) is consistent with a dust aerosol plume just above the PBL measured by the SLIM lidar (section 3.2). The $NO_2$ plume CAMS simulations (Fig.10) also show the advection of the low $O_3$ streamer located over Brittany and the English Channel on June 16 and east of Paris on June 17. The low $O_3$ layer measured by the DIAL above 1.5 km in Paris is indeed a regional feature not specific to the Paris city center. This southern advection above the PBL contributes also to the chemical composition of the PBL as there is a convergence of two streamers in the PES distribution below 1.2 km (Fig.9, Fig.S7 bottom row and S8 top row ). On June 18, there is no longer any difference in the PBL and the free troposphere circulation pattern (Fig.S4 middle row) in phase with the growing contribution of the dust plume in the PBL aerosol lidar backscatter (Fig.S4b) and the decrease of ozone in the PBL (Fig.7 and 5d).

- June 19 to 22: The origins of the air masses observed in Paris remain located in eastern France with limited long range advection both in the PBL and in the free troposphere. This is consistent with an aerosol plume of European continental pollution observed by the SLIM lidar on June 21 (Fig.S5a) and the advection of $NO_2$ continental plume and corresponding high $O_3$ concentrations from eastern to western France on June 22 (Fig.10). The low $NO_2$ concentrations east of the city centre in the CAMS simulation (Fig.10) also explain the positive differences observed on June 22 between the city center DIAL and the IAGOS in-situ observations (Fig.7) when the aircraft was flying east of Paris (Fig.S2).

- June 28: The influence of continental air masses is very limited on June 28 both in the PBL and in the free troposphere, while a well defined westerly flow controls the chemical composition according to the elevated PES values above the Atlantic Ocean (Fig.S9 bottom row). Such a circulation pattern explains both the elevated PBLH (no anticyclonic subsidence in the free troposphere) and $O_3$ concentrations less than 100 $\mu g.m^{-3}$despite the elevated surface temperature of Fig.2 on June 28 (advection of clean marine air masses).

- July 11-13: The horizontal advection shifts back to a northerly flow with transport of $O_3$ poor air from Benelux (Fig.4d) on July 11. This northerly flow is consistent with the Paris plume position in the southwestern part of Paris region on July 12 (Fig.4e). On July 13 the flow shifts back to westerly flow especially above the PBL bringing back the Paris plume above the city center and leading to high $O_3$ concentrations.

The positions of the Paris $O_3$ plume seen at 500 m in the CAMS simulations (Fig.3, 4) corresponds very well with the FLEXPART PES distribution below 1.2 km, showing that the long range transport analysis on previous days is consistent with the day to day variability of the city plume positions. The position of the city's plume on June 17, which is transported north at 18 UT (Fig.3d), is in fact still to the west of the city at 12 UT (not shown) and therefore remains consistent with the distribution of the FLEXPART PES of the city centrer area at 13 UT being maximum to the north and east of Paris (Fig.S4 top row).

## 4.3 Comparison of pollution episodes observed during ACROSS

Four pollution periods have been presented in the previous section. All 4 share conditions conducive to increasing $O_3$ concentrations in the lower troposphere above Paris: high temperatures (close to or above 30°C as shown in Fig.2), formation of an $O_3$ plume around Paris (see Fig.3,4), storage of $O_3$ concentrations photochemically produced during the day within a residual nocturnal layer (see Fig.5 to 6). The lowermost tropospheric $O_3$ columns also show extreme values above 13 DU for these 4 episodes (see orange rectangles in Fig.8). However, there are significant differences in the formation of regional-scale pollution plumes or the development of urban boundary layers over Paris to explain the variability of the extremes observed.

– June 14-18 case study: This period is characterized by a low PBLH < 1.5 km and advection of low $O_3$ and a dust plume in the free troposphere. Low $O_3$ concentrations have been frequently observed within dust plume in western Europe (Bonasoni et al., 2004; Andrey et al., 2014). Nevertheless the highest $O_3$ concentrations ($> 170\,\mu g.m^{-3}$) and lowermost tropospheric columns are found during this episode because European scale $O_3$ photochemical production took place in addition to local photochemistry in the Paris plume. Ozone pollution mitigation due to low $O_3$ concentrations in the dust plume took place only on June 18 when PBL and free troposphere mix more effectively.

– June 21-22 case study: The PBLH remains below 1.5 km while there is now an advection of continental plume with elevated $O_3$ ($> 140\,\mu g.m^{-3}$) and aerosol concentrations in the free troposphere. The frequent occurrence of clouds in the mid-troposphere and lower surface temperatures than during the first case study explain less $O_3$ within the Paris plume. The lowermost tropospheric columns are still above 14 DU because advection of the free tropospheric $O_3$ layer just above the PBL compensate lower $O_3$ production within the PBL.

– June 28 case study: Although the surface temperature is similar to the second case study, this pollution event is now characterized by elevated PBLH >2.5 km and no advection of continental plumes above the PBL. Only the Paris plume contributes to the $O_3$ photochemical production. This is consistent with $O_3$ concentrations $\leq 110\,\mu g.m^{-3}$ and lowermost tropospheric columns < 14 DU because $O_3$ photochemical precursors will be diluted over a greater thickness. Lower cloud cover than during the second case study is not sufficient to compensate for dilution of $O_3$ precursor emissions in the PBL.

– July 12-13 case study: The last pollution event is also characterized by elevated PBLH >2.5 km and no advection of a continental plume, even though surfaces temperatures are as high as during the first pollution event. This is why the lowermost tropospheric columns are again above 14 DU, but contrary to the second case study elevated $O_3$ concentrations $> 140\,\mu g.m^{-3}$ are mainly observed within the PBL.

Table 3 summarizes the main characteristics of the summer pollution episodes encountered in Paris. The diversity of long range transport and its role in $O_3$ variability means that this table can be considered sufficiently representative of the conditions that lead to a summer $O_3$ increase in a city like Paris. Three main conclusions can be drawn from our analysis:

**Table 3.** Characteristics of the Paris ozone episodes in summer 2022.

| Date | 14-18 June | 21-22 June | 28 June (or 2 July) | 11-13 July |
|---|---|---|---|---|
| $O_3$ plume altitude, km | <1.5 | <2.5 | <2.5 | <3 |
| $O_3$ plume maximum, $\mu$g.m$^{-3}$ | 170 | 150 | 110 | 150 |
| $O_3$ 0-3 km column, DU | 14-16 | 12-13 | 12 | 13-15 |
| High temperature, No clouds | Yes | No | No | Yes |
| PBL height maximum, km | 1.5 | 1.5 | 2.5 | 3.0 |
| PBL $O_3$ and $NO_2$ regional increase | Yes | Yes | No | 13 June only |
| Regional plume above PBL | Dust plume | European pollution | No | No |
| Bias IASI vs $O_3$ profiles, DU | -1.5 to -5 | 0 to -1.5 | -2 to -3.5 | 0 to -2 |

– Westward advection of the pollution plume from continental Europe enhance the $O_3$ increase over the city of Paris. The contribution of an increase in $O_3$ background has already been widely demonstrated for other megacities in North America, such as deep stratospheric intrusions or forest fire plumes (see next section). Deep stratospheric intrusions are rare from May to September in North Western Europe in comparison with North America (Akritidis et al., 2021). Long range transport of forest fire plumes are also detected in Europe, but at higher altitude (>5km) than in North America (Baars et al., 2021) with less contribution to the low troposphere $O_3$ background. Therefore westward advection of the pollution plume from continental Europe is a significant contribution for the Paris area.

– High temperatures in Paris are often accompanied by a southerly flow carrying Saharan dust in the 2-5 km altitude range over northern France (Israelevich et al., 2012). This study show that the downward entrainment of the low $O_3$ plume at the top of the polluted PBL must be accounted for to understand a possible mitigation of the PBL ozone increase during a summer heat wave.

– The maximum altitudes of the $O_3$ plume change from 1.5 km up to 3 km. The capability of IR satellite observations can be assessed using the ACROSS $O_3$ profile observations. Our study shows that IASI 0-3 km trophoperic $O_3$ column is sensitive to the day-to-day $O_3$ variability in the lower troposphere, especially when using the AM IASI observations. The significant underestimate of the 0-3 km partial column when the $O_3$ plume remains below 1.5 km, is reduced as soon as the plume maximum altitude exceeds 2 km.

## 4.4 Comparison with other works

LISTOS 2018-2019 and Southwestern USA campaigns took place in places and time periods which can be best compared with ACROSS, i.e. with limited fire and intercontinental pollution and STE. The main difference with LISTOS is the lack of

land-sea breeze recirculation for Paris. Ozone concentrations exceeded 200 $\mu$g.m$^{-3}$ during LISTOS with stagnation and land-sea breeze recirculation not seen during ACROSS (Couillard et al., 2021). The regional advection of European continental O$_3$ plume and of Saharan dust outbreak frequently associated to heat wave and pollution episode are also specific of the Paris area. Regarding the comparison with the TEXAQS and TRACER-AQ Southeastern USA campaigns, large O$_3$ concentrations > 200 $\mu$g.m$^{-3}$ are observed near Huston due to the contribution of numerous petrochemical plants in addition to the city emissions (Parrish et al., 2009; Senff et al., 2010), while such O$_3$ concentrations have never been reached during ACROSS. The same conclusion can be drawn from the comparison with the ESCOMPTE campaign O$_3$ observations when petrochemical plant and ship emission contributions to O$_3$ plume formation are comparable to the Houston area (Drobinski et al., 2007).

The O$_3$ long range transport observed during the Southwestern USA campaigns (CABOTS, LVOS) is different from the conditions encountered during ACROSS since STE, fire emission and Asian pollution plume transport significantly contributed to the O$_3$ inflow upstream of the local emission sources especially at altitudes above 2 km (Langford et al., 2022, 2017; Faloona et al., 2020). The latter makes difficult a direct comparison with the level of O$_3$ pollution encountered during ACROSS. The main similarity with the ACROSS results is the good agreement between the wide extension of the O$_3$ streamers shown by both the chemical transport models and the lidar and aircraft observations (Langford et al., 2022; Zhang et al., 2020). Indeed the CAMS model analysis during ACROSS are consistent with the O$_3$ observations presented in this paper and also show that the role of easterly flow from continental Europe replaces that played by the long range transport of fires and Asian pollution plumes during the Southwestern USA campaigns.

## 5  Conclusions

Four O$_3$ pollution events with surface concentrations above 100 $\mu$g.m$^{-3}$ and lowermost tropospheric columns greater than 14 DU have been encountered during the summer 2022 ACROSS campaign. In this work, vertical O$_3$ profiles measured by a UV DIAL, aircraft (IAGOS) and surface stations at different elevations in the Paris area have been analyzed in synergy with CAMS model simulations at different level in the lowermost troposphere, with PBL diurnal evolution using a 808-nm microlidar SLIM and radiosoundings and with FLEXPART simulations of the regional scale advection in the Paris PBL. The contribution of the DIAL lidar is essential to picture the role of the residual layer O$_3$ reservoir and that of advections of European continental pollution plumes or Saharan dust plumes above the boundary layer. We have shown in this study that the CAMS simulations of the Paris O$_3$ plume are consistent with the measurements of the O$_3$ vertical profiles and that the IASI satellite 0-3 km O$_3$ partial column day-to-day variability analysis benefits from vertical profile measurements. Satellite 0-3 km partial column with a significant negative bias can be flagged by looking at the maximum altitude level of the lower tropospheric O$_3$ plume. In addition to the well-known control of O$_3$ photochemical production in the urban plume by the surface temperature, by the cloud cover and by the mixing of the surface layer (0 - 500 m) with the residual layer, this work has shown that the thickness of the PBL during the day and the advection of regional scale plumes above the PBL can significantly change the O$_3$ concentrations. With similar cloud cover and air temperature, high O$_3$ concentrations up to 180 $\mu$g.m$^{-3}$ are encountered during the day when PBLH is below 1.5 km, while they remain below 150 $\mu$g.m$^{-3}$ when PBLH increases above 2.5 km. Advection of O$_3$ poor

concentrations in the free troposphere during a Saharan dust event is able to mitigate the $O_3$ photochemical production at the end of the first case study (June 18). On the other hand, the advection of a continental pollution plume with high $O_3$ concentrations $> 140$ μg.m$^{-3}$ maintained high concentrations in the surface layer despite a decrease in temperatures and an increase in cloud cover (June 22). Although the types of regional ozone plumes observed for pollution episodes in Paris are specific to the geographical position of this megacity, the need to take these regional contributions into account in order to understand the variability of pollution episodes in megacities is consistent with what has been observed in past campaigns. Regarding the interaction between the urban layer dynamical development and the $O_3$ plume formation during the day, this work is a preliminary study. Further analyses are needed to characterize this interaction in the lowermost troposphere around Paris using additional measurements of wind field and turbulent mixing, e.g. radar and Doppler lidar observations carried out during ACROSS. The microlidar observations will be also improved in the future to monitor continuously both the $O_3$ profile and the vertical structure of the atmospheric boundary layer. Finally the $O_3$ profiles presented in this paper in addition to aircraft chemical observations of the urban plume carried out during the 2022 ACROSS campaign onboard the French ATR-42 aircraft will be very valuable datasets to validate future mesoscale simulations of the formation and transport of the $O_3$ plume around Paris.

*Code and data availability.*

The IASI O3 products processed with FORLI-O3 are available at: http://iasi.aeris-data.fr/O3/, last access: 6 February 2024.

The AIRPARIF network $O_3$ data have downloaded from https://data-airparif-asso.opendata.arcgis.com/datasets/airparif-asso::2022-eiff3/ and https://data-airparif-asso.opendata.arcgis.com/datasets/airparif-asso::2022-pa13/

The QUALAIR station in-situ measurements ($O_3$, temperature) are available at http://qualair.aero.jussieu.fr/qualair.php?menu=\mhchem@cee{O3}&option=jussieu

The IAGOS have been downloaded from the IAGOS-AERIS web site https://iagos.aeris-data.fr/download/

The CAMS ENSEMBLE model hourly ANALYSIS of O3 concentration at 1 levels from 20220613-20220714 on Europe have been downloaded from the CAMS website https://ads.atmosphere.copernicus.eu/cdsapp#!/dataset/cams-europe-air-quality-forecasts?tab=overview

The DIAL data are available on the ACROSS campaign data base using the following keyword ACROSS-LATMOS-SU-QUALAIR-O3-profile-Lidar. The data base is hosted by the AERIS web site: https://across.aeris-data.fr/catalogue/

The SLIM lidar data are available at http://qualair.aero.jussieu.fr/

The radiosounding data are available at https://doi.org/10.25326/ and the skew-T diagrams have been plotted using the python library MetPy https://unidata.github.io/MetPy/latest/index.html

The Meteo France meteorological data for the Luxembourg and Tour Eiffel stations can be downloaded from https://meteo.data.gouv.fr/datasets/6569b51ae64326786e4e8e1a

The FLEXPART code version 9.2 was downloaded from the FLEXPART wiki homepage https://www.flexpart.eu/downloads and the meteorological analysis data extraction needed to run the FLEXPARt model have been carried out on the ECMWF ATOS data server using the flex-extract version 7.1.3 package downloaded from FLEXPART wiki homepage.

"ACROSS Ground Operation" National Programme to improve knowledge of chemical transformations in the atmosphere, the interaction between plant and human emissions, and their role on air quality.

*Author contributions.* G.Ancellet (GA) and F. Ravetta (FR) designed the work plan and are the PI of the DIAL. C. Viatte (CV) and C.Cailteau-Fischbach (CCF) provided the infrastucture of the QUALAIR station and CCF was responsible of the lidar deployment. CV and A. Boynard (AB) conducted the analysis of the lowermost tropospheric columns and of the IASI data. J. Pelon (JP) and Pascal Genau designed the SLIM lidar and conducted the analysis of the PBL structure. P. Nedelec (PN) provided the IAGOS data. Julie Capo (JC) and Axel Roy (AR) provided the meteorological soundings and contributed to the analysis the PBL dynamical development. GA processed the DIAL data and conducted the overall data synthesis. All contributed to the paper preparation.

*Competing interests.* No competing interest

*Acknowledgements.* The work was supported by Sorbonne Université and OSU Ecce TERRA through funding for running the QUALAIR Paris Station. IASI is a joint mission of EUMETSAT and the Centre National d'Etudes Spatiales (CNES, France). ULB-LATMOS is acknowledged for the development of the FORLI retrieval algorithm, and the AC SAF project of the EUMETSAT for providing IASI $O_3$ data.

The authors would like to acknowledge the QUALAIR team and infrastructure for their scientific support (https://qualair.fr/index.php/en/english/)

The PANAME experimental component benefits from supports from the H2C 4-year project funded by the French national agency for research (ANR) with the reference ANR-20-CE22-0013 and also from Météo-France and WMO as part of the Research Demonstration Project for Paris Olympics 2024 coordinated by Valéry Masson, and from Institut Pierre Simon Laplace supporting both measurements (SIRTA observatory and ACTRIS research infrastructure) and data management (AERIS national data and services center). The ACROSS ground based observations have been supported by the CNRS/INSU LEFE-CHAT project "ACROSS Ground Operation National Programme to improve knowledge of chemical transformations in the atmosphere, the interaction between plant and human emissions, and their role on air quality".

The authors acknowledge the AERIS data infrastructure for providing access to the IASI-FORLI data, and for hosting the IAGOS and ACROSS data base. The AIRPARIF Air Quality Agency is gratefully acknowledged for providing the $O_3$ surface data for Paris 13 and Eiffel tower stations. The European Centre for Medium Range Weather Forecasts (ECMWF) is acknowledged for the provision of meteorological analysis data and the FLEXPART development team is acknowledged for the provision of the FLEXPART 9.2 model version used in this publication.

Dr. Christopher Cantrell, PI of the ACROSS campaign is also gratefully acknowledged for his support.

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
