# Peer review of "Analysis of ozone vertical profiles day-to-day variability in the lower troposphere during the Paris-2022 ACROSS campaign"

_EGUsphere, 2024_

## Referee Comment (RC1)

The manuscript by *Ancellet et al.* (2024) analyzes results from the summer 2022 ACROSS (Atmospheric ChemistRy Of the Suburban foreSt) measurement campaign. This field campaign was conducted in, and around, the city of Paris and focused on observations of the diurnal and day-to-day variability of ozone ($O_3$) in the lower troposphere. Vertical profiles of atmospheric constituents were obtained from $O_3$ and aerosol lidars and commercial aircraft and were combined with radiosondes, IASI satellite retrievals, and CAMS model simulations to understand the processes driving spatiotemporal variability, vertical distributions, and magnitudes of $O_3$ in the planetary boundary layer (PBL). The manuscript presents in detail the physicochemical characteristics of numerous high $O_3$ events which occurred between June 13 and July 13, 2022. The text goes to great lengths to describe the agreement in different features (e.g., PBL and RL heights, $O_3$ concentrations, etc.) observed or simulated by the numerous measurement and modeling tool applied in this study. Four main $O_3$ events were intercompared for the physicochemical associated with the observed pollution values. The work highlights the importance of ground based $O_3$ lidars for better understanding air quality. I appreciated the effort the authors have gone through to provide all the details of results from the observations and modeling tools used during the campaign; however, the text does become dense at times. It would be nice if the authors could focus more on the main results of the study without discussing and intercomparing each observation/modeling data source for all 4 pollution events. Also, the novelty of this study is not immediately apparent. The manuscript is generally well-written; however, numerous typos were identified. Please see the minor and major comments below which I think would improve the overall manuscript.

**Minor Comments**

1. Line 1. "profile" not "profiles"

2. Line 10. "shows" not "show"

3. Line 48. "relative contribution *of*"…

4. Line 57. Is the last comma in this line supposed to be a period? The sentence between Line 55-60 needs some work. It is very hard to follow.

5. Line 62. The impact of long-range transport of $O_3$ has been shown in studies using ground-based lidar and satellites as well (e.g., Langford et al., 2019, 2022; Johnson et al., 2021).

6. Line 121. "remnants" instead of "remain".

7. Line 151. There is an extra ")".

8. Line 154. "The" Copernicus…; and "concentration" should be "concentrations".

9. Line 151. 10 km × 10 km

10. Line 156-157. Does the author mean "In this work, CAMS model analysis was conducted at 3 daily time steps…? This sentence needs some editing.

11. Line 157. October needs to be capitalized.

12. Line 202. "and" not "or".

13. Line 229. The authors start to use "$O_3$" for ozone about halfway through the paper. For consistency, it would be good to just use the chemical formula throughout the manuscript.

14. Line 158. I don't think you need "downloaded in october 2023" in this sentence. This information is better for the Data Availability section at the end of the manuscript.

15. Line 352. Missing a ")".

**Major Comments**

1. How were the IAGOS and lidar $O_3$ partial columns calculated? Was the IASI observational operator (averaging kernel and a priori profile) used to calculate these values from IAGOS and lidar data? Same question about the CAMS data shown in Fig. 13. This is an important step in order to have directly comparable information between satellite products and other observed/modeled data.

2. Figure 13. The authors compare IASI 0-3 km partial $O_3$ columns to IAGOS, lidar, and CAMS 0-3 km and 1.2-3 km partial $O_3$ columns. This figure shows that IASI 0-3 km partial $O_3$ columns are much lower compared to the IAGOS, lidar, and CAMS 0-3 km products; however, are more comparable to the 1.2-3 km partial $O_3$ columns from these three products. I am confused why the authors state this is such a good agreement. The IASI 0-3 km partial $O_3$ columns compared to IAGOS, lidar, and CAMS 0-3 km data suggests nearly a 100% underestimation by satellite data. The authors state that satellites have limited sensitivity to lower tropospheric $O_3$, which is true; however, the a priori information in the retrievals still exists. The limited sensitivity only limits the retrieval from deviating from the a priori state. The text reads as if the authors are saying the lowermost tropospheric $O_3$ values in the satellite retrievals will be near zero due to the limited sensitivity. Is this why the authors focus on the comparison of IASI 0-3 km partial $O_3$ columns to IAGOS, lidar, and CAMS 1.2-3 km partial $O_3$ columns? This is not correct.

3. Line 305-310. The authors are starting to touch on the true limitations of satellite sensitivity to lowermost tropospheric $O_3$ here; however, don't quite complete the statement. A main reason the satellite data agrees with observations on low $O_3$ days is that the a priori information for IASI is likely based on climatological information. Given the limited sensitivity of satellites to PBL pollution, the retrieval will result in values very similar to the a priori. The authors should expand upon this and reference the numerous studies that have been published on this.

4. For back-trajectory calculations, are there higher spatial resolution meteorological data that could be used to drive these simulations? 1° × 1° ECMWF meteorological data cannot capture the

city-scale features being observed during ACROSS. The entire domain shown in the supplemental figures only encompasses ~2 × 4 ECMWF grids.

5. This work highlights the importance of $O_3$ lidar data to better understand air quality and PBL dynamics throughout the day. It would be good to reference the many studies in the literature that have demonstrated this in the past especially those from observations made by the Tropospheric Ozone Lidar Network (TOLNet, https://tolnet.larc.nasa.gov/) (e.g., Langford et al., 2017, 2019, 2022; Sullivan et al., 2016, 2017; Johnson et al., 2021). Similar to the work here, these past studies, many conducted during large field campaigns, have shown the impact of local emissions, long-range transport of pollution, PBL heights, RL heights, meteorological conditions, and other physicochemical elements on local $O_3$ concentrations. These referenced works have focused on UV $O_3$ lidar observations, combined with ancillary observations and model simulations, to study nearly identical topics focused on in this work. It would be good for the authors to review these past studies and determine the similarities and differences between them and the work presented here by the authors.

6. The authors go through great lengths to discuss the physicochemical conditions observed and simulated during the ACROSS. However, the manuscript lacks discussion about what new has been found compared to past field campaigns and publications. The authors state at the end of the paper that "…interaction between the urban layer dynamical development and the $O_3$ plume formation during the day, this work is a first study". However, there are many studies which have discussed the impact of PBL/RL dynamics, local emissions, and long-range transport on observed $O_3$ formation. Just a small sample of these studies are referenced above. I think the authors could reduce the very lengthy text describing and intercomparing each observation/modeling tool for all four $O_3$ events in order to expand more on the novelty of this study. What new results were found during ACROSS? How does this advance the understanding of air quality? This needs to be discussed in detail because it is not clear to this reviewer that any novel findings were found. The authors should do a much more thorough literature review of this topic in order to identify the novelty of this work.

7. At times it feels there are too many figures in the paper. All 14 figures in the main text have multiple panels and become overwhelming. It would be easier for the reader if the authors focused their discussion on new findings and condense the figures in order to show the main results. The text is very dense when intercomparing every measurement and modeling tool for each case study. Perhaps the authors could improve the readability of the manuscript by only focusing on main findings instead of discussing every piece of information for every day throughout the campaign. At times it starts to read more like a field campaign report and less like a journal manuscript.

8. The final version of the paper should improve the quality of the figures. Some of the figures appear to have low resolution and some of the symbols used in them are not easy to see.

**References**

Johnson, M. S., Strawbridge, K., Knowland, K. E., Keller, C., and Travis, M.: Long-range transport of Siberian biomass burning emissions to North America during FIREX-AQ, Atmos. Environ., 252, 118241, https://doi.org/10.1016/j.atmosenv.2021.118241, 2021.

Langford, A. O., Alvarez, R. J., Brioude, J., Fine, R., Gustin, M. S., Lin, M. Y., Marchbanks, R. D., Pierce, R. B., Sandberg, S. P., Senff, C. J., Weickmann, A. M., and Williams, E. J.: Entrainment of stratospheric air and Asian pollution by the convective boundary layer in the southwestern U.S, J. Geophys. Res., 122, 1312–1337, https://doi.org/10.1002/2016JD025987, 2017.

Langford, A. O., Alvarez II, R. J., Kirgis, G., Senff, C. J., Caputi, D., Conley, S. A., Faloona, I. C., Iraci, L. T., Marrero, J. E., McNamara, M. E., Ryoo, J.-M., and Yates, E. L.: Intercomparison of lidar, aircraft, and surface ozone measurements in the San Joaquin Valley during the California Baseline Ozone Transport Study (CABOTS), Atmos. Meas. Tech., 12, 1889–1904, https://doi.org/10.5194/amt-12-1889-2019, 2019.

Langford, A. O., Senff, C. J., Alvarez II, R. J., Aikin, K. C., Baidar, S., Bonin, T. A., Brewer, W. A., Brioude, J., Brown, S. S., Burley, J. D., Caputi, D. J., Conley, S. A., Cullis, P. D., Decker, Z. C. J., Evan, S., Kirgis, G., Lin, M., Pagowski, M., Peischl, J., Petropavlovskikh, I., Pierce, R. B., Ryerson, T. B., Sandberg, S. P., Sterling, C. W., Weickmann, A. M., and Zhang, L.: The Fires, Asian, and Stratospheric Transport–Las Vegas Ozone Study (FAST-LVOS), Atmos. Chem. Phys., 22, 1707–1737, https://doi.org/10.5194/acp-22-1707-2022, 2022.

Sullivan, J. T., McGee, T. J., Langford, A. O., Alvarez, R. J., Senff, C. J., Reddy, P. J., Thompson, A. M., Twigg, L. W., Sumnicht, G. K., Lee, P., Weinheimer, A., Knote, C., Long, R. W., and Hoff, R. M.: Quantifying the contribution of thermally driven recirculation to a high-ozone event along the Colorado Front Range using lidar, J. Geophys. Res.-Atmos., 121, 10377–10390, https://doi.org/10.1002/2016JD025229, 2016.

Sullivan, J. T., Rabenhorst, S. D., Dreessen, J., McGee, T. J., Delgado, R., Twigg, L., and Sumnicht, G.: Lidar observations revealing transport of O3 in the presence of a nocturnal low-level jet: Regional implications for "next-day" pollution, Atmos. Environ., 158, 160–171, https://doi.org/10.1016/j.atmosenv.2017.03.039, 2017.

---

## Author Response (AR1)

**Answer to reviewer 1**

This document is the list of our responses to the reviewer's comments and a revised version of the text is also attached to this response to show the changes in red and the deleted sentences using strikethrough text

*The manuscript by Ancellet et al. (2024) analyzes results from the summer 2022 ACROSS (Atmospheric ChemistRy Of the Suburban foreSt) measurement campaign. This field campaign was conducted in, and around, the city of Paris and focused on observations of the diurnal and day-to-day variability of ozone (O3) in the lower troposphere. Vertical profiles of atmospheric constituents were obtained from O3 and aerosol lidars and commercial aircraft and were combined with radiosondes, IASI satellite retrievals, and CAMS model simulations to understand the processes driving spatiotemporal variability, vertical distributions, and magnitudes of O3 in the planetary boundary layer (PBL). The manuscript presents in detail the physicochemical characteristics of numerous high O3 events which occurred between June 13 and July 13, 2022. The text goes to great lengths to describe the agreement in different features (e.g., PBL and RL heights, O3 concentrations, etc.) observed or simulated by the numerous measurement and modeling tool applied in this study. Four main O3 events were intercompared for the physicochemical associated with the observed pollution values. The work highlights the importance of ground based O3 lidars for better understanding air quality. I appreciated the effort the authors have gone through to provide all the details of results from the observations and modeling tools used during the campaign; however, the text does become dense at times. It would be nice if the authors could focus more on the main results of the study without discussing and intercomparing each observation/modeling data source for all 4 pollution events. Also, the novelty of this study is not immediately apparent. The manuscript is generally well-written; however, numerous typos were identified. Please see the minor and major comments below which I think would improve the overall manuscript.*

We warmly thank the reviewer for his/her suggestions and comments.
In the introduction the objectives of the paper have been presented more explicitly with the following paragraph:
"The presentation of the O3 vertical observations available during this period as well as a preliminary analysis of the respective contribution of the urban boundary layer structure and of the O$_3$ plume regional transport are the main objectives of this paper. The latter has been extensively discussed for North American campaigns listed hereabove, but it is not clear if similar conclusions can be drawn for the Paris area about the role of elevated ozone concentrations transported from outside the megacity area. The Paris area is also different from the places with complicated pollution plume recirculation due to orography or land-sea breeze meteorological forcing where many previous campaigns took place in Europe or North America. Therefore it is relevant to present a study specific to the development of ozone pollution episode in the Paris area.
The overall description of the O$_3$ variability during the ACROSS campaign and the selection of the pollution events analyzed in this work are presented in section 3.1. This section focusses on lidar observations and their comparison with aircraft and model data. The comparison of the ACROSS O$_3$ vertical profiles and satellite observations, as well as a comparison of the pollution events in term of regional O$_3$ transport and PBL dynamical development are discussed in section 4. Section 4.1 first shows to what extent the O$_3$ measurements discussed in this work are relevant for studying the summer day-to-day variability of ozone in the lower troposphere in Paris, including the potential input from satellite observations. Section 4.2 presents the analysis of the regional O$_3$ transport during ACROSS since this process has been recognized during the past campaigns as a significant source of variability. Sections 4.3 and 4.4 summarize the main characteristics of the summer pollution episodes encountered during ACROSS and put the results into a broader perspective by comparing them with those of past measurement campaigns"
The structure of the paper has been modified to make the contribution of the work more readable with firstly a section 3 presenting the measurements discussed in the paper with fewer figures and more synthetic and with secondly a section 4 discussing the analysis of the results. We have

modified figures 5 to 12 (now figures 5 to 7) and have moved the microlidar data presentation in the supplementary document to focus on the ozone data analysis as requested by the reviewer. A summary table (Table 3) has been added to present the main characteristics of the summer pollution episodes encountered in Paris during ACROSS in section 4.3 and this section has been expanded to present the 3 main findings derived from this work. A new subsection 4.4 is added discussing similarities and differences with results obtained during past campaigns. A careful copy editing of English writing has been made.

*Minor Comments*
*1. Line 1. "profile" not "profiles"*
*2. Line 10. "shows" not "show"*
*3. Line 48. "relative contribution of"…*
*4. Line 57. Is the last comma in this line supposed to be a period? The sentence between Line 55-60 needs some work. It is very hard to follow.*
*5. Line 62. The impact of long-range transport of O3 has been shown in studies using ground-based lidar and satellites as well (e.g., Langford et al., 2019, 2022; Johnson et al., 2021).*
*6. Line 121. "remnants" instead of "remain".*
*7. Line 151. There is an extra ")".*
*8. Line 154. "The" Copernicus…; and "concentration" should be "concentrations". 9. Line 151. 10 km × 10 km*
*10. Line 156-157. Does the author mean "In this work, CAMS model analysis was conducted at 3 daily time steps…? This sentence needs some editing.*
*11. Line 157. October needs to be capitalized.*
*12. Line 202. "and" not "or".*
*13. Line 229. The authors start to use "O3" for ozone about halfway through the paper. For consistency, it would be good to just use the chemical formula throughout the manuscript.*
*14. Line 158. I don't think you need "downloaded in october 2023" in this sentence. This information is better for the Data Availability section at the end of the manuscript.*
*15. Line 352. Missing a ")".*

We thank the reviewer for his careful editing of the paper and all these minor corrections are included in the new version.

*Major Comments*

*1. How were the IAGOS and lidar O3 partial columns calculated? Was the IASI observational operator (averaging kernel and a priori profile) used to calculate these values from IAGOS and lidar data? Same question about the CAMS data shown in Fig. 13. This is an important step in order to have directly comparable information between satellite products and other observed/modeled data.*

In the revised manuscript, we have applied the IASI observational operator to the IAGOS, LIDAR and CAMS data. We have changed Figure 13 (new figure 8), to show both the raw and smoothed IAGOS, LIDAR and CAMS data. Finally, we have also modified Table 2 to directly compare raw and smoothed values of $O_3$ partial columns between IASI and the other observed/modeled data.

Table 2. Mean and standard deviation of $O_3$ 0-3km partial columns in Dobson Unit (DU) derived from raw and smoothed IAGOS, DIAL, and CAMS data, as well as IASI observations during the ACROSS campaign between June 13 to July 13 2022.

| | $O_3$ column (0 - 3 km DU) | | | | | | | |
|---|---|---|---|---|---|---|---|---|
| | raw | | | N | smoothed | | | N |
| IAGOS | 11.56 | ± | 1.93 | 49 | 8.53 | ± | 0.40 | 28 |
| DIAL | 12.88 | ± | 2.38 | 52 | 8.55 | ± | 0.49 | 42 |
| CAMS | 12.00 | ± | 1.77 | 32 | 7.83 | ± | 0.12 | 19 |

| | | | | | |
|---|---|---|---|---|---|
| **IASI AM** | 7.75 | ± | 1.37 | 19 | |
| **IASI PM** | 6.25 | ± | 0.98 | 19 | |
| **IASI** | 7.00 | ± | 1.40 | 38 | |

**We thank the reviewer for his careful editing of the paper and all these minor corrections are included in the new version.**

[Figure]

**Figure 8. Comparison of tropospheric lowermost O$_3$ column derived from the ACROSS observations (DIAL in blue and IAGOS in green), CAMS data (in red), and IASI satellite observations (morning – yellow diamonds, and evening – cyan diamonds) calculated in the [48.84°N- 49°N, 2°E-2.5°E] box between June 13 to July 13 2022. Circles and squares correspond to the 0-3km O$_3$ partial columns and smoothed partial columns, respectively. The orange boxes show the pollution days discussed in section 4.**

*2. Figure 13. The authors compare IASI 0-3 km partial O3 columns to IAGOS, lidar, and CAMS 0-3 km and 1.2-3 km partial O3 columns. This figure shows that IASI 0-3 km partial O3 columns are much lower compared to the IAGOS, lidar, and CAMS 0-3 km products; however, are more comparable to the 1.2-3 km partial O3 columns from these three products. I am confused why the authors state this is such a good agreement. The IASI 0-3 km partial O3 columns compared to IAGOS, lidar, and CAMS 0-3 km data suggests nearly a 100% underestimation by satellite data. The authors state that satellites have limited sensitivity to lower tropospheric O3, which is true; however, the a priori information in the retrievals still exists. The limited sensitivity only limits the retrieval from deviating from the a priori state. The text reads as if the authors are saying the lowermost tropospheric O3 values in the satellite retrievals will be near zero due to the limited sensitivity. Is this why the authors focus on the comparison of IASI 0-3 km partial O3 columns to IAGOS, lidar, and CAMS 1.2-3 km partial O3 columns? This is not correct.*

**We agree with the referee and we have removed the comparison with the 1.2-3 km O$_3$ partial columns in the revised manuscript. Instead, we have analyzed the sensitivity of the O$_3$ partial columns derived from IASI in terms of deviation from the a priori states, and Degrees Of Freedom for Signal (DOFS). Figure R1 shows that the O$_3$ 0-3 km partial columns and variabilities derived from IAGOS, DIAL and CAMS smoothed data are systematically lower than those calculated without taking into account the IASI averaging kernels. Figure R1 is only included in the answer to the reviewer. The following text has been added in section 4.1:**

**Smoothing with the IASI AKs reduces ozone columns and variability because part of the signal information comes from the *a priori* profile which is constant over time. However, IASI observations exhibit a variability of ~5 DU (mean of 7.00 ± 1.40 DU) over Paris during the ACROSS**

campaign, demonstrating that atmospheric signal is present in the retrieval information content with an averaged DOFS of 0.22 and 0.08 for morning and evening measurements, respectively.

[Figure]

**Figure R1: Timeseries of O$_3$ 0-3km partial columns of the retrievals (diamonds) and the *a priori* states (red dots), as well as Degrees Of Freedom for Signal (DOFS, squares) derived from IASI morning (yellow) and evening (cyan) observations.**

*3. Line 305-310. The authors are starting to touch on the true limitations of satellite sensitivity to lowermost tropospheric O3 here; however, don't quite complete the statement. A main reason the satellite data agrees with observations on low O3 days is that the a priori information for IASI is likely based on climatological information. Given the limited sensitivity of satellites to PBL pollution, the retrieval will result in values very similar to the a priori. The authors should expand upon this and reference the numerous studies that have been published on this.*

**Figure R1 above clearly show that IASI retrievals vary with time while the a priori column is constant over time. We show, in the new figure 8, that the day-to-day variability of IASI columns is of the same order of magnitude as that of O$_3$ IAGOS, DIAL and CAMS (5 DU). The following text has been also included in section 4.1:**
**IASI O$_3$ columns are overall lower than IAGOS, DIAL and CAMS raw and smoothed columns, with biases of the order of 1-3 DU, in particular when ozone partial columns above 2 km are low, such as between June 14$^{th}$ and 19$^{th}$, and between June 29$^{th}$ and July 5$^{th}$. Inversely, IASI and the smoothed IAGOS/DIAL O$_3$ columns are similar in the case of a high PBL (> 2.5 km) or in the case of high ozone above 2km (> 100 µg/m$^{-3}$), which are the cases on June 22th, June 28$^{th}$, and July 12$^{th}$.**

4. For back-trajectory calculations, are there higher spatial resolution meteorological data that could be used to drive these simulations? 1° × 1° ECMWF meteorological data cannot capture the city-scale features being observed during ACROSS. The entire domain shown in the supplemental figures only encompasses ~2 × 4 ECMWF grids.

**We agree with the reviewer that the resolution of the ECMWF used for the FLEXPART simulations may limit the analysis of city-scale features but these simulations are only used in section 4.2 focussing on the regional scale transport of the ozone plume. The final horizontal resolution of the FLEXPART simulation output product (here map of PES) is also smaller than the ECMWF grid as Lagrangian model are in principle independent of the initial wind horizontal resolution data (Stolh and Seibert, 1998; Stohl et al. 2002). We do not aim at using such simulations for a detailed description of the city-scale micrometeorological features. The city-scale ozone vertical vertical features are only discussed on the basis of the microlidar data**

and the Paris radiosoundings. Fig. 9 shows now an example of the output of a FLEXPART simulation in the main paper as requested by Reviewer 2 and the corresponding domain encompasses 10 x 25 ECMWF grids. This is good enough for our objective.

The following sentence has been included in section 4.2:

"The 1°x1° horizontal resolution of the ECMWF wind analysis is obviously limited for fine tracking of the city plume, but the PES FLEXPART distributions remain very accurate to check to what extent long range transport must be taken into account in the analysis of the city plume."

*5. This work highlights the importance of O3 lidar data to better understand air quality and PBL dynamics throughout the day. It would be good to reference the many studies in the literature that have demonstrated this in the past especially those from observations made by the Tropospheric Ozone Lidar Network (TOLNet, https://tolnet.larc.nasa.gov/) (e.g., Langford et al., 2017, 2019, 2022; Sullivan et al., 2016, 2017; Johnson et al., 2021). Similar to the work here, these past studies, many conducted during large field campaigns, have shown the impact of local emissions, long-range transport of pollution, PBL heights, RL heights, meteorological conditions, and other physicochemical elements on local O3 concentrations. These referenced works have focused on UV O3 lidar observations, combined with ancillary observations and model simulations, to study nearly identical topics focused on in this work. It would be good for the authors to review these past studies and determine the similarities and differences between them and the work presented here by the authors.*

We fully agree that the first version of the paper did not sufficiently detail the contribution of the numerous past campaigns, e.g. the results obtained in North America since the setup of the TOLNET network. We apologize for not having been explicit enough on this point, even if the previous introduction already recalled the numerous existing contributions on the role of processes controlling the intensity of pollution episodes.  The introduction has been updated with the following text:

"Several campaigns took place in North America to characterize high $O_3$ summer concentrations:  Texas Air Quality Study (TexAQS) 2000 and 2006 and TRacking Aerosol Convection ExpeRiment - Air Quality (TRACER-AQ) 2021 in Southwestern US (Daum 2004, Senff 2010, Liu 2023), California Research at the Nexus of Air Quality and Climate Change (CalNex), California Baseline Ozone Transport Study (CABOTS) 2016, Las Vegas Ozone Study (LVOS)  2016 and 2017 in California  (Ryerson2013, Langford2022, Faloona2020), Long Island Sound Tropospheric Ozone Study (LISTOS) 2018 and 2019 in New York City (Couillard 2021). During these campaigns extensive use of aircraft and lidar were conducted to better understand the sources and formation mechanism of $O_3$ plumes (Langford 2019). Results of LISTOS, CABOTS and TRACER-AQ show that meteorology and boundary layer heights are significant parameters influencing the vertical distribution of $O_3$ in these areas.  Sullivan (2017) demonstrated that residual $O_3$ layer reincorporation with mixed layer development contributes to a significant part of surface $O_3$ concentration increase in the afternoon. Contribution of long range transport of $O_3$ has been also analyzed using airborne differential absorption LIDAR (DIAL) and satellite. For example it was shown that regional transport of $O_3$ from Asian emissions over the North Pacific Ocean to California is responsible for a significant part of lower tropospheric $O_3$ increase in Summer (Lin2012, Langford2017) and that stratospheric-tropospheric exchanges (STE), forest fires and Asian pollution significantly control baseline ozone and therefore $O_3$ pollution in urban area in North America (Langford 2022, Wang 2021, Faloona 2020)."

A new section 4.4 is now devoted to comparing ACROSS results with those of previous campaigns, in particular those with the TOLNET network:

"LISTOS 2018-2019 and Southwestern USA campaigns took place in places and time periods which can be best compared with ACROSS, i.e. with limited fire and intercontinental pollution and STE. The main difference with LISTOS is the lack of land-sea breeze recirculation for Paris. Ozone concentrations exceeded 200 µg.m$^{-3}$ during LISTOS with stagnation and land-sea breeze recirculation not seen during ACROSS (Couillard et al., 2021). The regional advection of European continental O$_3$ plume and of Saharan dust outbreak frequently associated to heat wave and pollution episode are also specific of the Paris area. Regarding the comparison with the TEXAQS and TRACER-AQ Southeastern USA campaigns, large O$_3$ concentrations > 200 µg.m$^{-3}$ are observed near Huston due to the contribution of numerous petrochemical plants in addition to the city emissions (Parrish et al., 2009; Senff et al., 2010), while such O$_3$ concentrations have never been reached during ACROSS. The same conclusion can be drawn from the comparison with the ESCOMPTE campaign O$_3$ observations when petrochemical plant and ship emission contributions to O$_3$ plume formation are comparable to the Houston area (Drobinski et al., 2007). The O$_3$ long range transport observed during the Southwestern USA campaigns (CABOTS, LVOS) is different from the conditions encountered during ACROSS since STE, fire emission and Asian pollution plume transport significantly contributed to the O$_3$ inflow upstream of the local emission sources especially at altitudes above 2 km (Langford et al., 2022, 2017; Faloona et al., 2020). The latter makes difficult a direct comparison with the level of O$_3$ pollution encountered during ACROSS. The main similarity with the ACROSS results is the good agreement between the wide extension of the O$_3$ streamers shown by both the chemical transport models and the lidar and aircraft observations (Langford et al., 2022; Zhang et al., 2020). Indeed the CAMS model analysis during ACROSS are consistent with the O$_3$ observations presented in this paper and also show that the role of easterly flow from continental Europe replaces that played by the long range transport of fires and Asian pollution plumes during the Southwestern USA campaigns."

*6. The authors go through great lengths to discuss the physicochemical conditions observed and simulated during the ACROSS. However, the manuscript lacks discussion about what new has been found compared to past field campaigns and publications. The authors state at the end of the paper that "…interaction between the urban layer dynamical development and the O3 plume formation during the day, this work is a first study". However, there are many studies which have discussed the impact of PBL/RL dynamics, local emissions, and long-range transport on observed O3 formation. Just a small sample of these studies are referenced above. I think the authors could reduce the very lengthy text describing and intercomparing each observation/modeling tool for all four O3 events in order to expand more on the novelty of this study. What new results were found during ACROSS? How does this advance the understanding of air quality? This needs to be discussed in detail because it is not clear to this reviewer that any novel findings were found. The authors should do a much more thorough literature review of this topic in order to identify the novelty of this work.*

Again we apologize for not having been explicit enough on the high value of the results available from past campaigns. The use of the word "first study" and "first analysis" in the introduction and conclusion is a grammatical error made by a non-native English writer, we only maint that the paper is a preliminary analysis of the city-scale dynamical feature. This has been corrected.

As said earlier, section 3 has been significantly shortened to keep mainly the presentation of the ozone observations and the CAMS simulations. Section 4 has been expanded to summarize the main findings and add a new summary table (Table 3). The new version of section 4.3 now includes the following text:

"Table 3 summarizes the main characteristics of the summer pollution episodes encountered in Paris. The diversity of long range transport and its role in O$_3$ variability means that this table can be considered sufficiently representative of the conditions that lead to a summer O$_3$ increase in a city like Paris. Three main conclusions can be drawn from our analysis:

– Westward advection of the pollution plume from continental Europe enhance the $O_3$ increase over the city of Paris. The contribution of an increase in $O_3$ background has already been widely demonstrated for other megacities in North America, such as deep stratospheric intrusions or forest fire plumes (see next section). Deep stratospheric intrusions are rare from May to September in North Western Europe in comparison with North America (Akritidis et al., 2021). Long range transport of forest fire plumes are also detected in Europe, but at higher altitude (>5km) than in North America (Baars et al., 2021) with less contribution to the low troposphere $O_3$ background. Therefore westward advection of the pollution plume from continental Europe is a significant contribution for the Paris area.

– High temperatures in Paris are often accompanied by a southerly flow carrying Saharan dust in the 2-5 km altitude range over northern France (Israelevich et al., 2012). This study show that the downward entrainment of the low $O_3$ plume at the top of the polluted PBL must be accounted for to understand a possible mitigation of the PBL ozone increase during a summer heat wave.

– The maximum altitudes of the $O_3$ plume change from 1.5 km up to 3 km. The capability of IR satellite observations can be assessed using the ACROSS $O_3$ profile observations. Our study shows that IASI 0-3 km tropopheric $O_3$ column is sensitive to the day-to-day $O_3$ variability in the lower troposphere, especially when using the AM IASI observations.
The significant underestimate of the 0-3 km partial column when the $O_3$ plume remains below 1.5 km, is reduced as soon as the plume maximum altitude exceeds 2 km."

Table 3. Characteristics of the Paris ozone episodes in summer 2022.

| Date | 14-18 June | 21-22 June | 28 June (or 2 July) | 11-13 July |
|---|---|---|---|---|
| $O_3$ plume altitude, km | <1.5 | <2.5 | <2.5 | <3 |
| $O_3$ plume maximum, $\mu g.m^{-3}$ | 170 | 150 | 110 | 150 |
| $O_3$ 0-3 km column, DU | 14-16 | 12-13 | 12 | 13-15 |
| High temperature, No clouds | Yes | No | No | Yes |
| PBL height maximum, km | 1.5 | 1.5 | 2.5 | 3.0 |
| PBL $O_3$ and $NO_2$ regional increase | Yes | Yes | No | 13 June only |
| Regional plume above PBL | Dust plume | European pollution | No | No |
| Bias IASI vs $O_3$ profiles, DU | -1.5 to -5 | 0 to 1.5 | -2 to -3.5 | 0 to -2 |

*7. At times it feels there are too many figures in the paper. All 14 figures in the main text have multiple panels and become overwhelming. It would be easier for the reader if the authors focused their discussion on new findings and condense the figures in order to show the main results. The text is very dense when intercomparing every measurement and modeling tool for each case study. Perhaps the authors could improve the readability of the manuscript by only focusing on main findings instead of discussing every piece of information for every day throughout the campaign. At times it starts to read more like a field campaign report and less like a journal manuscript.*

As said earlier we strongly modified Fig. 5 to 14. There are now only 3 figures in section 3 (Fig. 5, 6, 7) to present the DIAL ozone data (including the height of the RL and PBL height as in the

first version). **The comparison between IAGOS, CAMS and DIAL vertical profiles are now shown in Fig. 7. We keep only the days where the comparison of IAGOS and DIAL is meaningful and we take into account only the lidar data that can be best compared with IAGOS (measurement times are now included in Fig. 7).**

[Figure]

**Figure 7. Daily mean O$_3$ vertical profiles in µg.m$^{-3}$ for the IAGOS aircraft (green) and the corresponding DIAL observations (blue) shown in Fig.5 to 6. Green times in UTC labeled within the figures are the IAGOS measurement times above Paris (two profiles per day except on June 14 and July 11). Blue times below the IAGOS flight times show the selection of the DIAL observations. CAMS model vertical profiles are also shown using horizontal averages of the model concentrations included in the Fig.1 area. CAMS profiles are shown at 6 UT (red □), 12 UT (red ∘) and 18 UT (red ▽).**

**We agree that the level of detail in the presentation of the different measurement days makes it more difficult to read the summary section 4. However, as in the numerous papers describing measurement campaigns, including those listed by the reviewer, it remains important to provide the reader with the information needed to contextualize the observations. We did our best to balance section 3 and 4 to show that the paper goes beyond a campaign report.**

*8. The final version of the paper should improve the quality of the figures. Some of the figures appear to have low resolution and some of the symbols used in them are not easy to see.*

**Fig. 5 to 7 have been changed to make them more readable.**

*References*
*Johnson, M. S., Strawbridge, K., Knowland, K. E., Keller, C., and Travis, M.: Long-range transport of Siberian biomass burning emissions to North America during FIREX-AQ, Atmos. Environ., 252, 118241, https://doi.org/10.1016/j.atmosenv.2021.118241, 2021.*
*Langford, A. O., Alvarez, R. J., Brioude, J., Fine, R., Gustin, M. S., Lin, M. Y., Marchbanks, R. D., Pierce, R. B., Sandberg, S. P., Senff, C. J., Weickmann, A. M., and Williams, E. J.: Entrainment of stratospheric air and Asian pollution by the convective boundary layer in the southwestern U.S, J. Geophys. Res., 122, 1312–1337, https://doi.org/10.1002/2016JD025987, 2017.*
*Langford, A. O., Alvarez II, R. J., Kirgis, G., Senff, C. J., Caputi, D., Conley, S. A., Faloona, I. C., Iraci, L. T., Marrero, J. E., McNamara, M. E., Ryoo, J.-M., and Yates, E. L.: Intercomparison of lidar, aircraft, and surface ozone measurements in the San Joaquin Valley during the California Baseline Ozone Transport Study (CABOTS), Atmos. Meas. Tech., 12, 1889–1904, https://doi.org/10.5194/amt-12-1889-2019, 2019.*
*Langford, A. O., Senff, C. J., Alvarez II, R. J., Aikin, K. C., Baidar, S., Bonin, T. A., Brewer, W. A., Brioude, J., Brown, S. S., Burley, J. D., Caputi, D. J., Conley, S. A., Cullis, P. D., Decker, Z. C. J., Evan, S., Kirgis, G., Lin, M., Pagowski, M., Peischl, J., Petropavlovskikh, I., Pierce, R. B., Ryerson, T. B., Sandberg, S. P., Sterling, C. W., Weickmann, A. M., and Zhang, L.: The Fires, Asian, and Stratospheric Transport–Las Vegas Ozone Study (FAST-LVOS), Atmos. Chem. Phys., 22, 1707–1737, https://doi.org/10.5194/acp-22-1707-2022, 2022.*
*Sullivan, J. T., McGee, T. J., Langford, A. O., Alvarez, R. J., Senff, C. J., Reddy, P. J., Thompson, A. M., Twigg, L. W., Sumnicht, G. K., Lee, P., Weinheimer, A., Knote, C., Long, R. W., and Hoff, R. M.: Quantifying the contribution of thermally driven recirculation to a high-ozone event along the Colorado Front Range using lidar, J. Geophys. Res.-Atmos., 121, 10377–10390, https://doi.org/10.1002/2016JD025229, 2016.*
*Sullivan, J. T., Rabenhorst, S. D., Dreessen, J., McGee, T. J., Delgado, R., Twigg, L., and Sumnicht, G.: Lidar observations revealing transport of O3 in the presence of a nocturnal low-level jet: Regional implications for "next-day" pollution, Atmos. Environ., 158, 160–171, https://doi.org/10.1016/j.atmosenv.2017.03.039, 2017.*

**This document is the list of our responses to the reviewer's comments and a revised version of the text is also attached to this response to show the changes in red and the deleted sentences using strikethrough text**

*Summary: DIAL Ozone profiles and IAGOS in situ data are presented during the 2022 ACROSS campaign on 21 days. These profiles are compared to the the satellite observations of Infrared Atmospheric Sounding Interferometer (IASI). Ancillary measurements from microlidar and radiosondes are also used for contextualizing the dynamics of the atmosphere. To better understand the regional transport of polluted air masses advected over the city, daily ozone analysis of the Copernicus Atmospheric Service (CAMS) ensemble model 10 and on backward trajectories of the Paris city plume were also utilized.*

*Major Comments: This paper aims to discuss the importance of DIAL profiles on understanding the pollution transport on several high ozone days during ACROSS 2022. This effort is unfortunately not very well documented or referenced and reads closer to a campaign report, rather than a scientifically significant manuscript.*

We warmly thank the reviewer for his/her suggestions and comments.
In the introduction the objectives of the paper have been presented more explicitly with the following paragraph:
"The presentation of the O3 vertical observations available during this period as well as a preliminary analysis of the respective contribution of the urban boundary layer structure and of the $O_3$ plume regional transport are the main objectives of this paper. The latter has been extensively discussed for North American campaigns listed hereabove, but it is not clear if similar conclusions can be drawn for the Paris area about the role of elevated ozone concentrations transported from outside the megacity area. The Paris area is also different from the places with complicated pollution plume recirculation due to orography or land-sea breeze meteorological forcing where many previous campaigns took place in Europe or North America. Therefore it is relevant to present a study specific to the development of ozone pollution episode in the Paris area.
The overall description of the $O_3$ variability during the ACROSS campaign and the selection of the pollution events analyzed in this work are presented in section 3.1. This section focusses on lidar observations and their comparison with aircraft and model data. The comparison of the ACROSS $O_3$ vertical profiles and satellite observations, as well as a comparison of the pollution events in term of regional $O_3$ transport and PBL dynamical development are discussed in section 4. Section 4.1 first shows to what extent the $O_3$ measurements discussed in this work are relevant for studying the summer day-to-day variability of ozone in the lower troposphere in Paris, including the potential input from satellite observations. Section 4.2 presents the analysis of the regional $O_3$ transport during ACROSS since this process has been recognized during the past campaigns as a significant source of variability. Sections 4.3 and 4.4 summarize the main characteristics of the summer pollution episodes encountered during ACROSS and put the results into a broader perspective by comparing them with those of past measurement campaigns"
The structure of the paper has been modified to make the contribution of the work more readable with firstly a section 3 presenting the measurements discussed in the paper with fewer figures and more synthetic and with secondly a section 4 discussing the analysis of the results. We have modified figures 5 to 12 (now figures 5 to 7) and have moved the microlidar data presentation in the supplementary document to focus on the ozone data analysis as requested by reviewer 1. A summary table (Table 3) has been added to present the main characteristics of the summer pollution episodes encountered in Paris during ACROSS in section 4.3 and this section has been expanded to present the 3 main findings derived from this work. A new subsection 4.4 is added discussing similarities and differences with results obtained during past campaigns. A careful copy editing of English writing has been made.

We agree that the level of detail in the presentation of the different measurement days makes difficult to emphasize the summary section 4. However, as in the numerous papers describing measurement campaigns, including those listed by the reviewer, it remains important to provide the reader with the information needed to contextualize the observations. We did our best to balance section 3 and 4 to show that the paper goes beyond a campaign report.

Section 4 has been expanded to summarize the main findings and add a new summary table (Table 3). The new version of section 4.3 now includes the following text:

"Table 3 summarizes the main characteristics of the summer pollution episodes encountered in Paris. The diversity of long range transport and its role in $O_3$ variability means that this table can be considered sufficiently representative of the conditions that lead to a summer $O_3$ increase in a city like Paris. Three main conclusions can be drawn from our analysis:

– Westward advection of the pollution plume from continental Europe enhance the $O_3$ increase over the city of Paris. The contribution of an increase in $O_3$ background has already been widely demonstrated for other megacities in North America, such as deep stratospheric intrusions or forest fire plumes (see next section). Deep stratospheric intrusions are rare from May to September in North Western Europe in comparison with North America (Akritidis et al., 2021). Long range transport of forest fire plumes are also detected in Europe, but at higher altitude (>5km) than in North America (Baars et al., 2021) with less contribution to the low troposphere $O_3$ background. Therefore westward advection of the pollution plume from continental Europe is a significant contribution for the Paris area.

– High temperatures in Paris are often accompanied by a southerly flow carrying Saharan dust in the 2-5 km altitude range over northern France (Israelevich et al., 2012). This study show that the downward entrainment of the low $O_3$ plume at the top of the polluted PBL must be accounted for to understand a possible mitigation of the PBL ozone increase during a summer heat wave.

– The maximum altitudes of the $O_3$ plume change from 1.5 km up to 3 km. The capability of IR satellite observations can be assessed using the ACROSS $O_3$ profile observations. Our study shows that IASI 0-3 km tropopheric $O_3$ column is sensitive to the day-to-day $O_3$ variability in the lower troposphere, especially when using the AM IASI observations. The significant underestimate of the 0-3 km partial column when the $O_3$ plume remains below 1.5 km, is reduced as soon as the plume maximum altitude exceeds 2 km."

*There is mention of pollution and ozone precursors, but the authors have failed to pull in any sort of additional chemical observations besides ozone. CAMS or IAGOS NOx or other species will help bolster the conclusions of pollution transport or why there are potentially differences between the measurements.*

The reviewer is right saying that there is no ozone precursor measurement included in this work. We tried to include the ACROSS ATR42 aircraft data in the paper, but there were not available for the days with elevated ozone pollution presented in this paper, except on June 22[nd]. However for this day the interesting feature is an ozone plume forming above 1.5 km, while the ATR42 flew at low level below 500 m (see Fig. R1 only included in this answer). The $NO_x$ plume observed by the ATR42 west of Paris below 500 m is consistent with the CAMS $NO_2$ simulation now shown in Fig. 10 west of Paris. We therefore choose to rely mainly on CAMS simulations to characterize the formation and transport of the ozone plume at the regional scale. This is also why we say in the introduction and conclusion that it is a preliminary study of the ozone pollution events encoutered during ACROSS and that additional data set and modelling dedicated to the ACROSS analysis must be considered in a future work.

[Figure]

**Fig. R1: ATR42 aircraft measurement of NO$_x$ in ug/m$^3$ horizontal distribution on June 22 from 12-14 UT at 400 m**

In addition to the CAMS O$_3$ simulations presented in section 3, a new figure  (Fig. 10) is added to show the CAMS NO$_2$ plumes distributions on June 16-17 and June 21-22 to strengthen the discussion about the regional plume transport in section 4.2. The purpose of this figure is to show the consequence of the June 16-17 advection of the Saharan plume and the June 21-22 advection of the Continental European plume on the NO$_2$ distribution and therefore the ozone photochemical production.

The following text is added then in section 4.2:

"The NO$_2$ plume CAMS simulations (Fig.10) also show the advection of the low O$_3$ streamer located over Brittany and the English Channel on June 16 and east of Paris on June 17. The low O$_3$ layer measured by the DIAL above 1.5 km in Paris is indeed a regional feature not specific to the Paris city center."

"This is consistent with an aerosol plume of European continental pollution observed by the SLIM lidar on June 21 (Fig.S5a) and the advection of NO$_2$ continental plume and corresponding high O$_3$ concentrations from eastern to western France on June 22 (Fig.10). The low NO$_2$ concentrations east of the city centrer in the CAMS simulation (Fig.10) also explain the positive differences observed on June 22 between the city center DIAL and the IAGOS in-situ observations (Fig.7) when the aircraft was flying east of Paris (Fig S2)"

[Figure]

**Fig. 10: CAMS ensemble mean NO$_2$ at 1000 m above Northern France on June 16 and 17 (top row) when dust plume advection at the PBL top is observed by the aerosol lidar and on June 21 and 22 (bottom row) when continental aerosol and O$_3$ plume advection at the PBL top is observed by both lidar. The orange star and dark-blue triangle are respectively the DIAL position and the CDG airport. The color scale is NO$_2$ concentration in ug.m$^{-3}$.**

*Furthermore, the IASI measurements are not carefully assessed, some work needs to be done in understanding the inherent value and uncertainty of these measurements.*

**In the revised manuscript, we have applied the IASI observational operator to the IAGOS, LIDAR and CAMS data. We have changed Figure 13 (new figure 8), to show both the raw and smoothed IAGOS, LIDAR and CAMS data. Finally, we have also modified Table 2 to directly compare raw and smoothed values of O$_3$ partial columns between IASI and the other observed/modeled data. The text of section 4.1 has been completely changed to discuss the new figure and Table. See also answer to reviewer 1 for more details.**

*Comments below are intended to help the paper form a more thorough conclusion.*

*Minor Comments: There is a lack of appropriate and topical references throughout most of the manuscript. References to previous air quality/ozone campaigns should be refreshed for more recent work, in addition to expanding to other megacities.*

**We fully agree that the first version of the paper did not sufficiently detail the contribution of the numerous past campaigns, e.g. the results obtained in North America since the setup of the TOLNET network. We apologize for not having been explicit enough on this point, even if the previous introduction already recalled the numerous existing contributions on the role of processes controlling the intensity of pollution episodes. The introduction has been updated with the following text:**

**"Several campaigns took place in North America to characterize high O$_3$ summer concentrations: Texas Air Quality Study (TexAQS) 2000 and 2006 and TRacking Aerosol**

Convection ExpeRiment - Air Quality (TRACER-AQ) 2021 in Southwestern USA (Daum 2004, Senff 2010, Liu 2023), California Research at the Nexus of Air Quality and Climate Change (CalNex), California Baseline Ozone Transport Study (CABOTS) 2016, Las Vegas Ozone Study (LVOS) 2016 and 2017 in California (Ryerson2013, Langford2022, Faloona2020), Long Island Sound Tropospheric Ozone Study (LISTOS) 2018 and 2019 in New York City (Couillard 2021). During these campaigns extensive use of aircraft and lidar were conducted to better understand the sources and formation mechanism of $O_3$ plumes (Langford 2019). Results of LISTOS, CABOTS and TRACER-AQ show that meteorology and boundary layer heights are significant parameters influencing the vertical distribution of $O_3$ in these areas. Sullivan (2017) demonstrated that residual $O_3$ layer reincorporation with mixed layer development contributes to a significant part of surface $O_3$ concentration increase in the afternoon. Contribution of long range transport of $O_3$ has been also analyzed using airborne differential absorption LIDAR (DIAL) and satellite. For example it was shown that regional transport of $O_3$ from Asian emissions over the North Pacific Ocean to California is responsible for a significant part of lower tropospheric $O_3$ increase in Summer (Lin2012, Langford2017) and that stratospheric-tropospheric exchanges (STE), forest fires and Asian pollution significantly control baseline ozone and therefore $O_3$ pollution in urban area in North America (Langford 2022, Wang 2021, Faloona 2020)."

A new section 4.4 is now devoted to comparing ACROSS results with those of previous campaigns, in particular those with the TOLNET network:

"LISTOS 2018-2019 and Southwestern USA campaigns took place in places and time periods which can be best compared with ACROSS, i.e. with limited fire and intercontinental pollution and STE. The main difference with LISTOS is the lack of land-sea breeze recirculation for Paris. Ozone concentrations exceeded 200 µg.m$^{-3}$ during LISTOS with stagnation and land-sea breeze recirculation not seen during ACROSS (Couillard et al., 2021). The regional advection of European continental $O_3$ plume and of Saharan dust outbreak frequently associated to heat wave and pollution episode are also specific of the Paris area. Regarding the comparison with the TEXAQS and TRACER-AQ Southeastern USA campaigns, large $O_3$ concentrations > 200 µg.m$^{-3}$ are observed near Huston due to the contribution of numerous petrochemical plants in addition to the city emissions (Parrish et al., 2009; Senff et al., 2010), while such $O_3$ concentrations have never been reached during ACROSS. The same conclusion can be drawn from the comparison with the ESCOMPTE campaign $O_3$ observations when petrochemical plant and ship emission contributions to $O_3$ plume formation are comparable to the Houston area (Drobinski et al., 2007). The $O_3$ long range transport observed during the Southwestern USA campaigns (CABOTS, LVOS) is different from the conditions encountered during ACROSS since STE, fire emission and Asian pollution plume transport significantly contributed to the O3 inflow upstream of the local emission sources especially at altitudes above 2 km (Langford et al., 2022, 2017; Faloona et al., 2020). The latter makes difficult a direct comparison with the level of $O_3$ pollution encountered during ACROSS. The main similarity with the ACROSS results is the good agreement between the wide extension of the $O_3$ streamers shown by both the chemical transport models and the lidar and aircraft observations (Langford et al., 2022; Zhang et al., 2020). Indeed the CAMS model analysis during ACROSS are consistent with the $O_3$ observations presented in this paper and also show that the role of easterly flow from continental Europe replaces that played by the long range transport of fires and Asian pollution plumes during the Southwestern USA campaigns."

*L90 – Reference needed as to where this statement can be drawn from "The accuracy of the lidar observations is altitude-dependent being of the order of 7µg.m−3 below 1000 m and occasionally increases up to 20 µg.m−3 above 2 km at midday". Also recommend adding in a percentage difference. Please also note somewhere the conversion to ppbv for these observations - 1 ppb O3 = 1.96 µg/m³ at 25°C and 1 atm*

**Done**

*Fig. 3 – Higher resolution terrain maps in the background would help better understand the ozone transport throughout the time. Adding in the wind barbs would also contextualize which direction the plume was moving.*

**We did not included terrain map as orography is not an issue for studying the Paris area. We cannot easily produce a wind representation corresponding to the CAMS simulations in Fig. 3 and 4. These two figures with multiple panels contain already a lot of information and there is a dedicated section (section 4.2) dedicated to the analysis of the regional transport based on FLEXPART simulations**

*Fig 6b – Is the CBLH actually over 3.5-3.8km? This seems unrealistic, even with >30C temperatures. Is this an aged polluted air mass that has recirculated associated with the synoptic high pressure system over the area as mentioned in the text. This figure should be clarified or manual inspection of the the CLBH algorithm should be addressed. How did CAMS compare in terms of the RL and CLBH observations?*

**We agree that June 18 is an unsual event of CBLH growth over Paris, especially considering the time of the CBLH maximum (20:30 UT). All the CBLH calculations shown in this paper have been manually checked. We are also confident with this value as the radiosounding inversion layer was also at 3.5 km at 20 UT on this day. Also surface temperature was 38 °C on this day (Fig. 2). We anyway do not want to focus our paper too much on this interesting case in term of PBL dynamical development because $O_3$ DIAL observations after 15 UT are not available and $O_3$ was decreasing at the surface on June 18[th] because of the pollution mitigation by the dust plume advection over Northern France.**

*Fig 9 – It's unclear where and when these IAGOS data overlap. For instance on 20220615, what is the coincidence in time for the CAMS (or IAGOS) and DIAL?*

**The direct comparison between IAGOS, CAMS and DIAL vertical profiles is now better shown in the new section 3 (Fig. 7). We keep only the days where the comparison of IAGOS and DIAL is meaningful using daily mean and we take into account only the lidar data that can be best compared with IAGOS (measurement times are now included in Fig. 7). On June 15 there is a single IAGOS flight at 14 UT will the DIAL data are missing from 10-15 UT, therefore the comparison with IAGOS is not considered anymore in Fig. 7.**

[Figure]

**Figure 7.** Daily mean O$_3$ vertical profiles in µg.m$^{-3}$ for the IAGOS aircraft (green) and the corresponding DIAL observations (blue) shown in Fig.5 to 6. Green times in UTC labeled within the figures are the IAGOS measurement times above Paris (two profiles per day except on June 14 and July 11). Blue times below the IAGOS flight times show the selection of the DIAL observations. CAMS model vertical profiles are also shown using horizontal averages of the model concentrations included in the Fig.1 area. CAMS profiles are shown at 6 UT (red □), 12 UT (red ∘) and 18 UT (red ▽).

*Table 2 – are the +/- associated with the variance of the dataset or uncertainty associated with the observations? The relative levels of uncertainty between high precision DIAL and in-situ observations needs to be described in comparison to the likely much higher uncertainty satellite observations.*

The ± reflects the 1-sigma standard deviation around the mean for all dataset, not the uncertainties. In the revised manuscript, we have applied the IASI observational operators to the

IAGOS, LIDAR and CAMS data in order to take into account the differing characteristics of the observing systems, particularly their averaging kernels and error covariances of the satellite observations (Rodgers and Connor, 2003). We have also modified Table 2 to directly compare raw and smoothed values of $O_3$ partial columns between IASI and the other observed/modeled data.

Table 2. Mean and standard deviation of $O_3$ 0-3km partial columns in Dobson Unit (DU) derived from raw and smoothed IAGOS, DIAL, and CAMS data, as well as IASI observations during the ACROSS campaign between June 13 to July 13 2022.

| | $O_3$ column (0 - 3 km DU) | | | | | | | |
|---|---|---|---|---|---|---|---|---|
| | raw | | | N | smoothed | | | N |
| IAGOS | 11.56 | ± | 1.93 | 49 | 8.53 | ± | 0.40 | 28 |
| DIAL | 12.88 | ± | 2.38 | 52 | 8.55 | ± | 0.49 | 42 |
| CAMS | 12.00 | ± | 1.77 | 32 | 7.83 | ± | 0.12 | 19 |
| IASI AM | 7.75 | ± | 1.37 | 19 | | | | |
| IASI PM | 6.25 | ± | 0.98 | 19 | | | | |
| IASI | 7.00 | ± | 1.40 | 38 | | | | |

Reference: Rodgers, C. D., and B. J. Connor (2003), Intercomparison of remote sounding instruments, *J. Geophys. Res.*, 108, 4116, doi:10.1029/2002JD002299, D3.

*L305 – This statement regarding excellent agreement cannot be fully stated until the uncertainty estimations are presented or some level of description of the apriori data for IASI is described. References are critically needed throughout this section.*

We agree with the referee and we have removed the comparison with the 1.2-3km $O_3$ partial columns in the revised manuscript. Instead, we have analyzed the sensitivity of the $O_3$ partial columns derived from IASI in terms of deviation from the a priori state, and Degrees Of Freedom for Signal (DOFS). Figure R2 (only included in this answer) shows that the $O_3$ 0-3 km partial columns and variabilities derived from IAGOS, DIAL and CAMS smoothed data are systematically lower than those calculated without taking into account the IASI averaging kernels. Smoothing with the IASI AKs reduces ozone columns and variability because part of the signal information comes from the *a priori* profile which is constant over time. However, IASI observations exhibit a variability of ~5 DU (mean of 7.00 ± 1.40 DU) over Paris during the ACROSS campaign, demonstrating that atmospheric signal is present in the retrieval information content with an averaged DOFS of 0.22 and 0.08 for morning and evening measurements, respectively.

[Figure]

**Figure R2: Timeseries of O$_3$ 0-3km partial columns of the retrievals (diamonds) and the a priori states (red dots), as well as Degrees Of Freedom for Signal (DOFS, squares) derived from IASI morning (yellow) and evening (cyan) observations.**

*Figure 13 – The IAGOS data does not replicate some of the higher ozone concentrations as observed in the DIAL measurements. What is the reason for this? This should also be labeled Partial Ozone columns in the x-axis.*

**The IAGOS profiles are only twice a day with the first profile in early morning at 4-6 UT and the second one either at 10 UT or at 14 UT. DIAL data are generally available each day for a longer time period 6 UT to 20 UT (see Fig. 5 and 6). The ozone maximun being observed in the afternoon it is not suprising to observe the largest variability with the DIAL data. The day-to-day variability is anyway still visible in the IAGOS data in Figure 8. As said earlier, the direct comparison between IAGOS, CAMS and DIAL vertical profiles is now better shown in the new section 3 (Fig. 7).**

**We have modified Figure 13 (new Figure 8) with the new y-label.**

*Section 5.2  - This could be better visualized by bringing at least one of the FLEXPART simulation plots to the main paper rather than the supplemental.*

**Done**

---

## Author Response (AR2)

We apologize for not following the abstract length requipement listed in the preparation guidelines. The new abstract has been halved to stay under the 250-word limit.

Abstract :
The ozone vertical profiles variability in the Paris area is analyzed using 21 days of lidar monitoring of the lower troposphere ozone vertical profiles and planetary boundary layer (PBL) vertical structure evolution in summer 2022. Characterization of the pollution regional transport is based on daily ozone analysis of the Copernicus Atmospheric Service (CAMS) ensemble model and on backward trajectories. The CAMS simulations of the ozone plume between the surface and 3 km are consistent with the ozone measurements. Comparisons with tropospheric ozone column retrieved by the satellite observations of Infrared Atmospheric Sounding Interferometer (IASI) show that IASI observations can capture the day to day variability of the 0-3 km  ozone column only when the maximum altitude of the ozone plume is higher than 2 km.
The lidar ozone vertical structure above the city center is also in good agreement with the PBL growth during the day and with the formation of the residual layer during the night. The analysis of four ozone pollution events shows that the thickness of the PBL during the day and the advection of regional scale plumes above the PBL can significantly change the ozone concentrations above Paris.  Advection of ozone poor concentrations in the free troposphere during a Saharan dust event is able to mitigate the ozone photochemical production. On the other hand, the advection of a pollution plume from continental Europe with high ozone concentrations > 140 $\mu g.m^{-3}$ maintained high concentrations in the surface layer despite a temperature decrease and cloud cover development.